# A distributed differential game approach to trajectory planning for offshore wind farm inspection

Yunqi Liao[1], Shuyuan You[2], Houmin Wang[2], Siming Yu[1], Wenyan Xue[1]*

1 The College of Mechanical and Energy Engineering, Guangdong Ocean University, Yangjiang, China,
2 The College of Computer Science and Engineering, Guangdong Ocean University, Yangjiang, China

* wyxuedu@163.com

## Abstract

To address the complex challenges associated with multiple unmanned aerial vehicles (multi-UAVs) cooperative inspection in offshore wind farms, including limited sensing and communication ranges, constrained battery capacity, and round-trip mission requirements, this paper introduces an optimal coordinated trajectory method for multi-UAV based on a distributed differential game (DDG) framework. The approach explicitly accounts for energy consumption, incorporating round-trip requirements into a game-theoretic objective function to facilitate energy-aware trajectory planning. Each UAV operates based solely on local information from neighboring UAVs, enabling distributed decision-making that ensures collision-free coordination while optimizing global inspection time and overall energy efficiency. The convergence of the proposed strategy to a global Nash equilibrium (G-NE), as confirmed by theoretical analysis, ensures system-level coordination optimality subject to round-trip and energy constraints. Simulation results demonstrate that the method significantly enhances inspection efficiency and reduces task completion time by up to 18.7% compared to conventional approaches, while guaranteeing the safe return of all UAVs.

## Introduction

The field of offshore wind turbine inspection has witnessed a paradigm shift, moving from conventional methods toward intelligent solutions powered by unmanned aerial vehicles (UAVs) [1,2]. Early trajectory planning methods were largely guided by a distance-based nearest-first principle, which disregarded essential environmental factors such as direction and consequently led to poor energy efficiency and suboptimal inspection outcomes [3]. To address this limitation, a value evaluation function incorporating parameters such as positional altitude, average wind speed, and wind direction, along with an improved consensus-based bundle algorithm, is introduced, by which the rationality of trajectory planning was markedly enhanced [4]. However, this

**Data availability statement:** All relevant data for this study are within the paper and publicly available from the GitHub repository: (https://github.com/xuewenyan6-debug/Wenyan-Xue).

**Funding:** This work is supported by the funding from the Efficient space-time coordination of swarm aircraft (No. 360302022401, Funded Author: Wenyan Xue); The Zhanjiang Non-funded Science and Technology Research Project (2025B01076, Funded Author: Wenyan Xue). The Research on Motion Control Mechanism and Regulation Strategy for Brain Computer Interface (No. 360302042406, Funded Author: Houmin Wang). The 2025 University Student Innovation Training Program (CXXL2025259, Funded Author: Yunqi Liao).

model does not account for the energy constraint associated with the UAVs' return flight [5]. To reduce energy consumption in multi-UAVs, Ref. [6] introduced a genetic algorithm-based dynamic zoning strategy (GA-DZ), which optimizes UAV trajectories by minimizing the total flight distance, thereby implicitly enhancing energy efficiency. However, this approach suffers from limited adaptability and does not account for the impact of battery power on the return journey [7]. To address these limitations, multi-agent reinforcement learning has emerged as a promising alternative. For example, Ref. [8] applied such methods to enable agents to learn cooperative policies through extensive environmental interactions. Similarly, Ref. [9] combined convolutional neural networks with deep reinforcement learning (NN-DRL), creating an interactive mechanism between environmental perception and policy learning that considerably increased mission completion rates. Nevertheless, the challenge of aligning local optimization with global efficiency remains only partially resolved [10]. Distributed optimization methods, such as based on consensus algorithm [11], have been proposed to alleviate reliance on a central coordinator, yet it often require iterative communication and may not explicitly account for dynamic collision avoidance or energy constraints in time-critical missions. Furthermore, centralized optimization techniques, including mixed-integer linear programming, can generate optimal trajectories by solving a global optimization problem [12]. Nonetheless, their inherent dependency on a central coordinator and perfect global information makes them vulnerable to single points of failure and communication bottlenecks, which are common challenges in offshore settings. Recent efforts have explored hybrid centralized-distributed architectures [13] to balance optimality and robustness, but the fundamental issue of guaranteeing global performance with strictly local interactions persists. While effective in mitigating coordination conflicts, these approaches are often validated empirically, lacking the rigorous theoretical guarantees needed to provide proofs of optimality [14]. Alternatively, distributed model predictive control (DMPC) has been applied to handle unexpected changes in dynamic environments [15]. It synthesizes a state-feedback control strategy using a receding-horizon scheme. While DMPC can iteratively seek locally optimal solutions at each sampling instant, its focus on algorithmic optimization often prioritizes certain global objectives at the expense of individual UAV performance, which may lead to extended overall mission durations.

To bridge this gap, differential game theory has emerged as a promising framework for modeling multi-agent strategic interactions under dynamics constraints. It has been applied to various multi-robot coordination problems, such as formation control [16] and airborne conflict resolution [17]. This framework reconciles individual and collective objectives from a game equilibrium perspective, providing a theoretical foundation for analyzing system-level outcomes. Specific to distributed settings, recent work has investigated graphical differential games for networked systems with limited communication [18,19], establishing convergence to local Nash equilibria (L-NE) under certain connectivity conditions. Building upon this, Refs. [20,21] advanced multi-player differential game solutions via a framework merging distributed optimal control with game theory, specifically addressing collision avoidance.

However, despite these advances, the direct application of existing differential game formulations to optimal trajectory planning for offshore wind turbine inspection poses distinct challenges, particularly under the stringent constraints of limited sensing, communication, and most critically, finite energy for round-trip missions. The main obstacles are as follows: 1) The NE derived in many generic multi-agent games or even in existing offshore wind inspection scenarios do not effectively enhance operational efficiency for battery-constrained UAVs, as round-trip energy constraints are seldom incorporated into the cost function design [22,23]. 2) Many theoretical differential game solutions assume perfect or periodic global information exchange [24], an assumption often invalid in practical offshore wind scenarios due to limited and unreliable communication links that restrict information exchange to a local neighborhood. The issue of scalability and performance under imperfect communication becomes more pronounced as the number of UAVs increases and operational conditions grow more complex [25].

In summary, while GA-DZ [6] optimizes flight distance (and thus implicitly reduces energy consumption) yet lacks explicit safety and return-trip constraints, and NN-DRL [9] learns adaptive policies but offers no guarantee of global optimality and is prone to local optima, DMPC [15] can handle dynamic disruptions but often sacrifices inspection efficiency due to its local optimization nature. Furthermore, existing centralized differential game approaches [19,24] are not directly applicable to offshore wind farm inspection scenarios under communication constraints.

To overcome the above limitations, this paper proposes a novel DDG method, which provides a theoretically guaranteed globally optimal solution. The key contributions are summarized as follows:

1) Compared to the GA-DZ method [6], which requires re-iteration and thus suffers from reduced real-time performance when dealing with dynamic maritime environments, and which optimizes for distance while lacking return-trip constraints, the proposed DDG framework explicitly models round-trip energy constraints and local communication limitations, thereby significantly improving task completion efficiency while ensuring safety. and global convergence.

2) Unlike the NN-DRL approach [9], which learns adaptive strategies yet cannot ensure global optimality and is prone to local optima, the proposed DDG framework provides a theoretically guaranteed convergence from a L-NE to a G-NE for all UAVs. This overcomes the key limitations of learning-based strategies, particularly their lack of theoretical interpretability and convergence assurance.

3) In contrast to the DMPC [15], which relies on iterative algorithmic optimization to obtain locally optimal solutions at each sampling instant, the proposed DDG method is grounded in game theory and explicitly models strategic interactions among UAVs, driving the system toward a NE. Simulations further confirm the superior inspection efficiency of the proposed DDG method over prevailing trajectory planning methods in offshore wind farm applications [6,9,15].

## Preliminaries

A comprehensive list of variables and parameters is provided in Table 1.

## The problem description for offshore wind power inspection

### The description of inspection

The workflow for inspecting offshore wind turbines using a multi-UAV system is illustrated below [26]:

1. Task allocation: Inspection tasks are formulated by the control center based on a comprehensive assessment of turbine conditions and meteorological information, distributing them via a cloud platform to specify detection targets and priorities.

2. Coordinated control and data collection: During autonomous flight and data collection, UAVs operate within a coordinated control framework that harmonizes global objectives (thorough inspection of wind turbine components) with local

**Table 1. Nomenclature.**

| | |
|---|---|
| $p_{ix}$ (Eq.(1)) | The positional component along the x axis of UAV i |
| $p_{iy}$ (Eq.(1)) | The positional component along the y axis of UAV i |
| $p_{iz}$ (Eq.(1)) | The positional component along the z axis of UAV i |
| $\varepsilon_i$ (Eq.(1)) | The roll angle of UAV i |
| $\varsigma_i$ (Eq.(1)) | The pitch angle of UAV i |
| $\vartheta_i$ (Eq.(1)) | The yaw angle of UAV i |
| $m_i$ (Eq.(1)) | The mass of the UAV i |
| $I_{i_{xx}}$ (Eq.(1)) | The moment of inertia along the x axis of UAV i |
| $I_{i_{yy}}$ (Eq.(1)) | The moment of inertia along the y axis of UAV i |
| $I_{i_{zz}}$ (Eq.(1)) | The moment of inertia along the z axis of UAV i |
| $L_i$ (Eq.(1)) | The distance between the motor axis and the center of the body for UAV i |
| $g_i$ (Eq.(1)) | The acceleration due to gravity of UAV i |
| $\hat{b}_i$ (Eq.(2)) | The lift coefficient of UAV i |
| $\hat{d}_i$ (Eq.(2)) | The drag coefficient of UAV i |
| $\bar{p}_i(t)$ (Eq.(3)) | The position of UAV i at time t |
| $p_i(t)$ (Eq.(3)) | The pose of UAV i at time t |
| $u_i(t)$ (Eq.(3)) | The control strategy of UAV i at time t |
| $v_i(t)$ (Eq.(3)) | The velocity of UAV i at time t |
| $\mathcal{U}$ (Eq.(3)) | The all admissible control input set |
| $\bar{a}_i$ (Eq.(3)) | The velocity gain matrix of UAV i |
| $\bar{b}_i$ (Eq.(3)) | The control gain matrix of UAV i |
| $a_i$ (Eq.(4)) | The state gain matrix of UAV i |
| $A$ (Eq.(5)) | The state gain matrix of the UAVSs |
| $z$ (Eq.(5)) | The state of the UAVSs |
| $z_i^d$ (Eq.(6)) | The state of the target wind turbine for inspection |
| $z_i$ (Eq.(6)) | The state of UAV i |
| $\tilde{z}_i$ (Eq.(6)) | The state error of UAV i |
| $\mathcal{N}_i$ | The set of neighbors of UAV i |
| $u_i^*, u_{-i}^*$ (Eq.(8)) | The optimal strategy of UAV i and the neighbors, respectively |
| $z_i(t_f)$ (Eq.(9)) | The terminal state of the UAV i |
| $z_{ib}$ (Eq.(9)) | The base position terminal state of the UAV i |
| $u_{ij}$ (Eq.(9)) | The control strategy of neighbor UAV j |
| $\bar{R}_i$ (Eq.(11)) | The visual radius of UAV i |
| $\hat{\mathcal{O}}_i$ (Eq.(11)) | All known static obstacles as well as unknown obstacles within the sensing range of UAV i |
| $\bar{R}_{\bar{n}}$ (Eq.(11)) | The radius of obstacle $\bar{n}$ |
| $\hat{o}_{\bar{n}}$ (Eq.(11)) | The centroid of obstacle $\bar{n}$ |
| $\gamma_i$ (Eq.(12)) | The obstacle avoidance angle |
| $v_{i_{\hat{x}\hat{y}}}^x$ (Eq.(12)) | The velocity decomposed along the $\hat{x}$ in the $\hat{X}-\hat{O}-\hat{Y}$ coordinate |
| $v_{i_{\hat{x}\hat{y}}}^y$ (Eq.(12)) | The velocity decomposed along the $\hat{y}$ in the $\hat{X}-\hat{O}-\hat{Y}$ coordinate |
| $v_{i_{xy}}^x$ (Eq.(12)) | The velocity decomposed along the x in the $X-O-Y$ coordinate |

*(Continued)*

**Table 1.** (Continued)

| | |
|---|---|
| $v_{i_{xy}}^x$ (Eq.(12)) | The velocity decomposed along the $y$ in the $X-O-Y$ coordinate |
| $v_{o_{i_n}}$(Eq.(13)) | The obstacle avoidance velocity |
| $\hat{\gamma}_i$(Eq.(15)) | The deviation angle of UAV $i$ |
| $\hat{R}_i$ | The safety radius between UAV $i$ and obstacles |

goals (collision avoidance and inter-UAV safety). Under constraints including limited communication range, and energy capacity, trajectory planning is optimized to minimize task completion time while ensuring complete and accurate data acquisition.

3. Data processing and alerting: Collected data is transmitted in real-time for AI-based analysis to identify anomalies. Alerts are generated and pushed to maintenance terminals for rapid decision-making.

4. Return and recharging: After task completion, UAVs autonomously return to base, execute precise landing, automatically recharge, and backup data for subsequent missions.

**Remark 1.** *This study focuses on collaborative control of UAV clusters, excluding subsequent maintenance processes.*

## The modelling of UAV

This paper employs a quadrotor model to address the inspection of offshore wind turbines [27,28].

$$
\begin{cases}
\ddot{p}_{ix} & = \left(-\sin\varepsilon_i\sin\vartheta_i - \cos\varepsilon_i\sin\varsigma_i\cos\vartheta_i\frac{u_{i1}}{m_i}\right), \\
\ddot{p}_{iy} & = \left(-\cos\varepsilon_i\sin\varsigma_i\sin\vartheta_i + \sin\varepsilon_i\cos\vartheta_i\frac{u_{i1}}{m_i}\right), \\
\ddot{p}_{iz} & = -\cos\varepsilon_i\cos\varsigma_i\frac{u_{i1}}{m_i} + g_i, \\
\ddot{\varepsilon}_i & = \frac{u_{i2}L_i}{I_{ixx}} + \dot{\varsigma}_i\dot{\vartheta}_i\frac{I_{iyy}-I_{izz}}{I_{ixx}}, \\
\ddot{\varsigma}_i & = \frac{u_{i3}L_i}{I_{iyy}} + \dot{\varepsilon}_i\dot{\vartheta}_i\frac{I_{izz}-I_{ixx}}{I_{iyy}}, \\
\ddot{\vartheta}_i & = \frac{u_{i4}L_i}{I_{izz}} + \dot{\varepsilon}_i\dot{\varsigma}_i\frac{I_{ixx}-I_{iyy}}{I_{izz}},
\end{cases}
\tag{1}
$$

where $p_{ix}$, $p_{iy}$ and $p_{iz}$ are the positional components along the $x$, $y$, and $z$ axes on the 3-dimensional Euclidean space, respectively; $\varepsilon_i$, $\varsigma_i$ and $\vartheta_i$ are the roll, pitch, and yaw angles, respectively; $m_i$ is the mass of the UAV $i$; $I_{ixx}$, $I_{iyy}$ and $I_{izz}$ are the moments of inertia along the $x$, $y$, and $z$ axes, respectively; $L_i$ is the distance between the motor axis and the center of the body; $g_i$ is the acceleration due to gravity; $u_{i1}$, $u_{i2}$, $u_{i3}$, $u_{i4}$ are the control strategies of the UAV $i$, defined as follows:

$$
\begin{bmatrix} u_{i1} \\ u_{i2} \\ u_{i3} \\ u_{i4} \end{bmatrix} =
\begin{bmatrix}
\hat{b}_i & \hat{b}_i & \hat{b}_i & \hat{b}_i \\
0 & \hat{b}_i & 0 & -\hat{b}_i \\
\hat{b}_i & 0 & -\hat{b}_i & 0 \\
\hat{d}_i & -\hat{d}_i & \hat{d}_i & -\hat{d}_i
\end{bmatrix}
\begin{bmatrix} \hat{\omega}_{i1}^2 \\ \hat{\omega}_{i2}^2 \\ \hat{\omega}_{i3}^2 \\ \hat{\omega}_{i4}^2 \end{bmatrix},
\tag{2}
$$

where $\hat{b}_i$ is the lift coefficient of the UAV $i$; $\hat{d}_i$ is the drag coefficient; $\hat{\omega}_{i1}$, $\hat{\omega}_{i2}$, $\hat{\omega}_{i3}$ and $\hat{\omega}_{i4}$ are the rotation angular velocity of rotor 1, 2, 3, and 4 for the UAV $i$, respectively; $u_{i1}$ is the total vertical thrust; $u_{i2}$ is the differential lift affecting the pitch motion of the UAV $i$; $u_{i3}$ is the differential lift affecting the roll motion of the UAV $i$; $u_{i4}$ is the torque affecting the yaw motion of the UAV $i$.

 

To streamline the coordinated control design, the following assumption is introduced.

**Assumption 1.** *Each UAV operates with slow dynamics and small attitude angles near its equilibrium point, implying that the terms $\varepsilon_i$ and $\varsigma_i$ are negligible and can be approximated as zero.*

Under Assumption 1, the model for UAV $i(\forall i \in \mathbb{N}_{1:N})$ is defined with the control input $u_i(t) = [u_{i1}, u_{i2}, u_{i3}, u_{i4}]^T$ acting on the position $\bar{p}_i(t) = [p_{ix}, p_{iy}, p_{iz}]^T$ and yaw $\vartheta_i$. Consequently, the system model reduces to a second-order integrator dynamics.

For each UAV $i$ in the set $\mathbb{N}_{1:N}$, the model is

$$\begin{cases} \dot{p}_i(t) = \bar{a}_i v_i(t), \\ \dot{v}_i(t) = \bar{b}_i u_i(t) + \bar{g}_i, \end{cases} \tag{3}$$

where $p_i(t) = [\bar{p}_i(t)^T, \vartheta_i]^T \in \mathbb{R}^4$ is the pose of UAV $i$; $v_i(t) \in \mathbb{R}^4$ is the velocity of UAV $i$; $u_i(t) \in \mathcal{U} \subset \mathbb{R}^4$ is the control strategy of UAV $i$; $\mathcal{U}$ is the all allowable control input; $\bar{g}_i = [g, 0, 0, 0]^T \in \mathbb{R}^4$; $\bar{a}_i = I_4$; $\bar{b}_i = diag\{-\frac{1}{m_i}, 0, 0, \frac{1}{I_{zz,i}}\} \in \mathbb{R}^{4\times4}$

Then, the dynamics of each UAV are expressed by the following model:

$$\dot{z}_i = a_i z_i + b_{ii} u_i + \hat{g}_{ii}, \tag{4}$$

where $z_i = [p_i^T, v_i^T] \in \mathbb{R}^8$ is the state of UAV $i$; $a_i = \begin{bmatrix} 0_4 & \bar{a}_i \\ 0_4 & 0_4 \end{bmatrix} \in \mathbb{R}^{8\times8}$; $b_{ii} = \begin{bmatrix} 0_4 \\ \bar{b}_i \end{bmatrix} \in \mathbb{R}^{8\times4}$; $\hat{g}_{ii} = [0_{4\times1}^T, g, 0_{3\times1}^T]^T \in \mathbb{R}^8$.

Define the collective state of the multi-UAV system as $z = [z_1^T, \cdots, z_i^T, \cdots, z_N^T]^T \in \mathbb{R}^{8N}$. The resulting system dynamics are given by:

$$\dot{z} = Az + \sum_{i=1}^{N} \left( B_i u_i + \hat{g}_i \right), \tag{5}$$

where $B_i = [0, \cdots, 1, \cdots, 0]^T \otimes \begin{bmatrix} 0_4 \\ \bar{b}_i \end{bmatrix} \in \mathbb{R}^{8N\times4}$; $A = \begin{bmatrix} 0_4 & \bar{a}_i \\ 0_4 & 0_4 \end{bmatrix} \otimes I_N \in \mathbb{R}^{8N\times8N}$; $\hat{g}_i = [0, \cdots, 1, \cdots, 0]^T \otimes \hat{g}_{ii} \in \mathbb{R}^{8N}$.

To quantify the inspection deviation, we define the state error of UAV $i$ as:

$$\tilde{z}_i = z_i - z_i^d, \tag{6}$$

where $\tilde{z}_i(t) \in \mathbb{R}^{8N}$; $z_i^d \in \mathbb{R}^{8N}$ is the state of the target wind turbine for inspection.

Let $\tilde{z} = [\tilde{z}_1^T, \cdots, \tilde{z}_i^T, \cdots, \tilde{z}_N^T]^T \in \mathbb{R}^{8N}$ denote the collective state error vector of the multi-UAV system. The control objective is therefore formulated as driving this error to zero asymptotically, ensuring each UAV converges to its target wind turbine:

$$\lim_{t\to\infty} \tilde{z} = 0. \tag{7}$$

To characterize limited sensing and communication in the multi-UAV inspection system, the communication relationships are modeled using graph theory. Specifically, a directed graph $G(\mathcal{V}, \varepsilon)$ characterizes the topology among $N$ UAVs: $\mathcal{V} = \{1, 2, \ldots, N\}$ denotes the set of vertices, while $\varepsilon \subset \mathcal{V} \times \mathcal{V}$ denotes the set of edges representing communication links [29]. The presence of an edge $e_{ij} = e_{ji} \in \varepsilon$ indicates that UAV $i$ obtains information from UAV $j$. Accordingly, the neighbors of UAV $i$ are defined as $\mathcal{N}_i = \{j \in \mathcal{V} : (j, i) \in \varepsilon, j \neq i\}$. This paper assumes that the communication topology of the multi-UAV inspection system is directed and strongly connected.

## Problem statement for optimal multi-UAV coordination

During the execution of offshore wind turbine inspection tasks, multi-UAVs inherently encounter challenges including communication constraints, structural obstacles from the turbines, and potential inter-UAV conflicts. Consequently, the

trajectory planning problem can be effectively transformed into a coordinated control framework, which is essential for ensuring that all UAVs complete their inspection missions safely and efficiently.

To formulate the trajectory planning problem as a coordination control framework, we project the operational environment, including UAVs, turbines, and obstacles, into the configuration space. In line with the conventions established in Refs. [20,22], we define collision regions $\hat{\mathcal{S}}_o$, sensing regions $\hat{\mathcal{S}}_i$ and free regions $\hat{\mathcal{S}}_f$, representing non-navigable areas, collision avoidance areas and safe flight spaces, respectively (see [20,22] for details). Consistent with most UAV control studies, we give the the following assumptions:

**Assumption 2.** *For every UAV $i(i \in \mathbb{N}_{1:N})$, neither its initial position nor its target position lies within the collision region $\hat{\mathcal{S}}_o$.*

**Problem 1.** *(Distributed differential game) Consider a inspection system composed of N UAVs, whose dynamics are limited by the constraints of equations (4)-(6), and they operate in an environment containing collision areas $\hat{\mathcal{S}}_o$ (which include static obstacles such as wind turbine towers and blades, as well as other UAVs). The system is also subject to a communication topology $G(\mathcal{V}, \varepsilon)$, and round-trip requirements. The objective is to minimize the total task completion time while ensuring operational safety. This problem can be formulated within a DDG framework, where each UAV is treated as an intelligent player.*

*In this DDG framework, each player designs its coordinated control strategy $u_i$ to minimize an individual cost function $J_i(z(0))$, the specific form of which is deferred. Strategic interactions among player i and the neighbors $j(j \in \mathcal{N}_i)$ lead to a L-NE, which defines a collectively optimal control strategy and thus yields the optimal trajectory plan. Then,*

$$J_i^* \left( z(0), u_i^*, u_{-i}^* \right) \le J_i \left( z(0), u_i, u_{-i}^* \right) . \tag{8}$$

*where the pair $\left( u_i^*, J_i^* \left( z(0), u_i^*, u_{-i}^* \right) \right)$ denotes the optimal strategy and its associated cost for UAV i, while $u_{-i}^*$ represents the collective optimal strategy for its neighbors.*

## The model of DDG

In the context of offshore wind turbine inspection, the proposed DDG framework is shown in Fig 1. Each UAV *i* operates as an autonomous player. It receives local sensor data (own state $z_i$) and communicated information from neighbors $j \in \mathcal{N}_i$ (states $z_{ij}$, strategies $u_{ij}$). These inputs feed into its local Game Solver (green block), which solves the TPBVP (Eqs. (20)-(21)) via the numerical method (i.e., distributed dradient optimization for L-NE) to compute its optimal strategy $u_i^*$. This strategy is applied to its dynamics and also broadcast to its neighbors, closing the distributed feedback loop. The cyan block indicates information flow limited by the communication graph $G(\mathcal{V}, \varepsilon)$.

Specifically, the cost function for each UAV is defined as:

$$J_i \left( z(0), u_i^*, u_{-i}^* \right) \triangleq \psi(z_i(t_f)) + \int_0^{t_f} \left( \| Y_i(z_i) \|_{Q_i}^2 + \| u_i(t) \|_{R_i}^2 + \sum_{j \in \mathcal{N}_i} \| u_{ij}(t) \|_{R_{ij}}^2 \right) dt. \tag{9}$$

s.t.

$$\begin{cases} v_i \in [0, v_{max}], \\ u_i \in [u_{min}, u_{max}]. \end{cases}$$

where $F_i, Q_i, R_i, R_{ij}$ are are symmetric positive definite matrices; $\psi(z_i(t_f)) = \| z_i(t_f) - z_{ib} \|_{F_i}^2$ is the terminal return cost; $z_i(t_f)$ is the terminal state of the UAV *i*; $z_{ib}$ is the base position of UAV *i* (the takeoff position); $Y_i(z_i)$ denotes the running cost (to be defined explicitly later); and $u_{ij}$ corresponds to the control strategy of neighboring UAV $j(j \in \mathcal{N}_i)$.

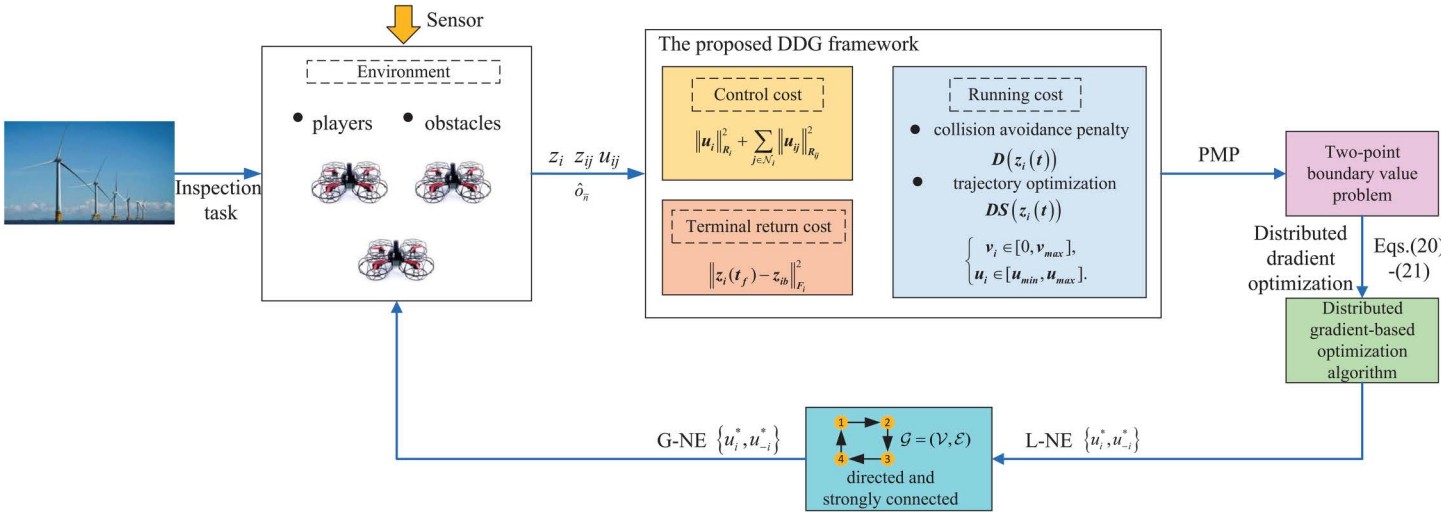

**Fig 1. The schematic diagram of the proposed DDG framework.**

The running cost for UAV $i$ is given by:

$$Y_i\left(z(t)\right) = \hat{\alpha}_1 D(z_i(t)) + \hat{\alpha}_2 DS(z_i(t)).$$

(10)

where $\hat{\alpha}_1 (0 < \hat{\alpha}_1 < 1)$, $\hat{\alpha}_2 (0 < \hat{\alpha}_2 < 1)$ are the weighting coefffcients; The collision avoidance penalty function, $D(z_i(t))$ is defined as:

$$D(z_i(t)) \triangleq \begin{cases} \infty & \text{for} \quad i \in \hat{\mathcal{S}}_o \\ \sum_{\bar{n}=1}^{|\hat{\mathcal{O}}_i|} \dfrac{\left((\bar{R}_i+\bar{R}_{\bar{n}})^2-\|p_i-\hat{o}_{\bar{n}}\|^2\right)^2}{\left(\|p_i-\hat{o}_{\bar{n}}\|^2-\left(\bar{R}_i+\hat{R}_{\bar{n}}\right)^2\right)^2} \|\hat{v}_i\|^2 & \text{for} \quad i \in \hat{\mathcal{S}}_i \\ 0 & \text{for} \quad i \in \hat{\mathcal{S}}_f \end{cases} ,$$

(11)

where the visual radius of UAV $i$ is denoted by $\bar{R}_i$, and $\hat{\mathcal{O}}_i$ represents the set of all known static obstacles along with unknown obstacles—including other UAVs and any unknown static obstacles within UAV $i$'s sensing range. For each obstacle $\bar{n}$, its radius and centroid are given by $\bar{R}_{\bar{n}}$ and $\hat{o}_{\bar{n}}$, respectively. $\hat{o}_{\bar{n}} = [o_{\bar{n}}^T, \vartheta_{\bar{n}}]^T$; $\vartheta_{\bar{n}} = 0$ if obstacle $\bar{n}$ is static, and $\vartheta_{\bar{n}} \neq 0$ if it is a UAV. The relative velocity $\hat{v}_i = v_{i_{xy}} - v_{\hat{o}_{\bar{n}}} \in \mathbb{R}^2$ is defined as the difference between the velocity of UAV $i$ projected onto the $X - Y$ plane, denoted as $v_{i_{xy}} \in \mathbb{R}^2$, and the obstacle avoidance velocity $v_{o_{\bar{n}}} \in \mathbb{R}^2$, which will be specified subsequently.

Let $X - O - Y$ and $\hat{X} - \hat{O} - \hat{Y}$ represent the world and body-fixed coordinate frames, respectively. The flight trajectory of UAV $i$ is projected onto the two-dimensional $X - Y$ plane. The velocity of UAV $i$ expressed in the body-fixed frame $\hat{X} - \hat{O} - \hat{Y}$ is defined as $v_{i_{\hat{x}\hat{y}}} = [v_{i_{\hat{x}\hat{y}}}^x, v_{i_{\hat{x}\hat{y}}}^y]^T \in \mathbb{R}^2$, representing its components along the $\hat{X}$ and $\hat{Y}$ axes. This velocity vector can be obtained by

$$\begin{bmatrix} v_{i_{\hat{x}\hat{y}}}^x \\ v_{i_{\hat{x}\hat{y}}}^y \end{bmatrix} = \begin{bmatrix} \cos\gamma_i & -\sin\gamma_i \\ \sin\gamma_i & \cos\gamma_i \end{bmatrix} \begin{bmatrix} v_{i_{xy}}^x \\ v_{i_{xy}}^y \end{bmatrix},$$

(12)

where $\gamma_i$ denotes the obstacle avoidance angle. The terms $v_{i_{\hat{x}\hat{y}}}^x$ and $v_{i_{\hat{x}\hat{y}}}^y$ correspond to the velocity projections of UAV $i$ onto the $\hat{x}$ and $\hat{y}$ axes of its body-fixed frame $\hat{X}-\hat{O}-\hat{Y}$, respectively. Similarly, $v_{i_{xy}}^x$ and $v_{i_{xy}}^y$ are its velocity projections onto the $x$ and $y$ axes of the world coordinate system $X-O-Y$.

Accordingly, the obstacle avoidance velocity is denoted by $v_{\hat{O}_{\bar{n}}}$ and is defined as:

$$v_{\hat{O}_{\bar{n}}} = \begin{bmatrix} \cos\gamma_i & \sin\gamma_i \\ \sin\gamma_i & \cos\gamma_i \end{bmatrix}^{-1} \begin{bmatrix} v_{i_{\hat{x}\hat{y}}}^x \\ v_{i_{\hat{x}\hat{y}}}^y \end{bmatrix}. \tag{13}$$

Fig 2 illustrates the obstacle avoidance angle, denoted as $\gamma_i$

$$\gamma_i = \begin{cases} \hat{\bar{\aleph}}_i + \arcsin\frac{\bar{R}_{\bar{n}}}{\|\hat{o}_{\bar{n}}-p_i\|} & \text{for} \quad \bar{n} \in \mathcal{O}^{kn} \\ \hat{\bar{\aleph}}_i + \arcsin\frac{\hat{R}_i}{\|\bar{o}_{\bar{n}}-p_i\|} & \text{for} \quad \bar{n} \in \mathcal{O}^{un} \end{cases}, \tag{14}$$

where $\hat{\bar{\aleph}}_i = \arccos\frac{\hat{o}_{\bar{n}}^x-p_i^x}{\|\hat{o}_{\bar{n}}-p_i\|}$; $\mathcal{O}^{kn}$ and $\mathcal{O}^{un}$ are the sets of known static obstacles and unknown obstacles, respectively; $\hat{o}_{\bar{n}}^x$ and $p_i^x$ are the positions decomposed along the $x$ axis in the $X-O-Y$ coordinate.

**Remark 2.** *In the paper, when UAV i is avoiding an unknown obstacle, the obstacle centroid used in the proposed obstacle penalty function refers to the centroid based on the boundary points of the obstacle detected within the sensing range of the UAV i. The radius of the obstacle used is determined by the UAV's safety radius $\hat{R}_i$.*

$DS(z_i(t))$ is the trajectory optimization function, which is designed as:

$$DS(z_i) = \begin{cases} \infty & \text{for} \quad i \in \hat{\mathcal{S}}_o, \\ \frac{\|\arccos\hat{\gamma}\|}{\pi} & \text{for} \quad otherwise. \end{cases} \tag{15}$$

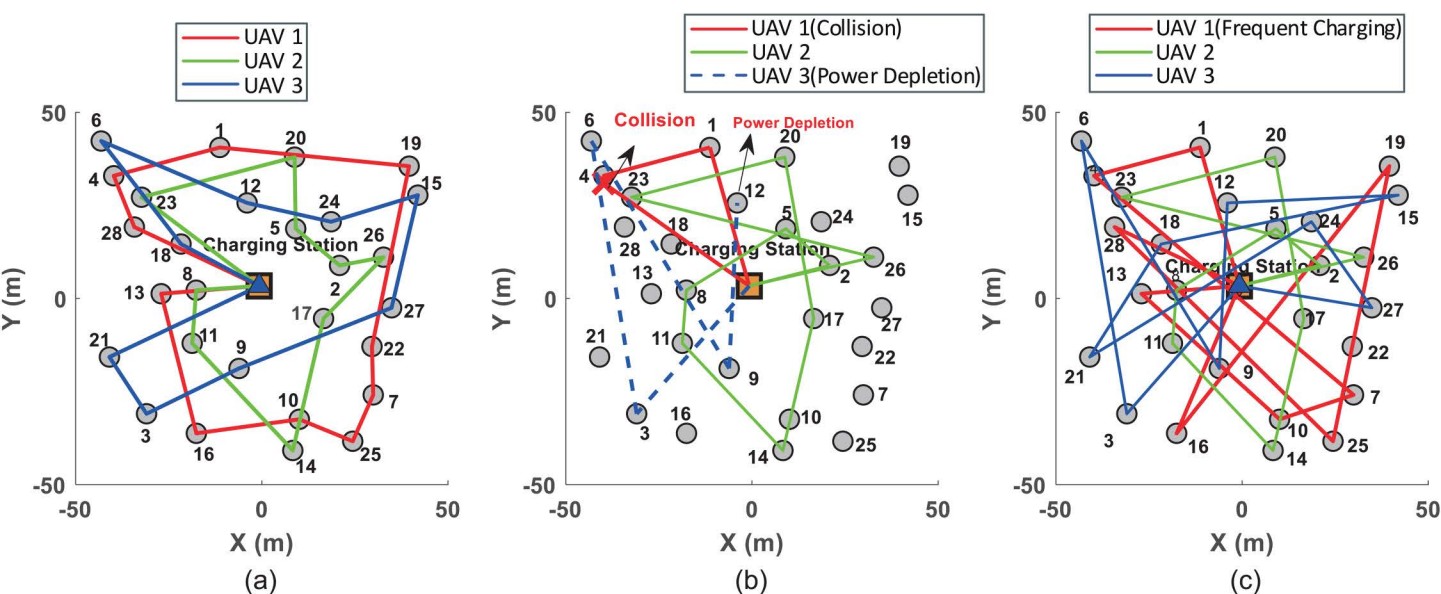

**Fig 2. Comparison of inspection trajectories for 3 UAVs and 28 turbines. (a)** Proposed DDG method: All UAVs (UAV1-3) complete inspection tasks. **(b)** GA-DZ method [6]: UAV1 collides with Wind Turbine 4 (marked by 'X'), and UAV3 exhausts its battery before returning. **(c)** NN-DRL method [9]: UAV1 exhibits an inefficient, elongated path requiring mid-mission returns to the OBS.

where $\hat{\gamma}$ is the deviation angle of UAV $i$. $\cos\hat{\gamma} = \frac{<(z_i(0)-z_i^d),(z_i(0)-\tilde{z}_i-z_i^d)>}{\|z_i(0)-z_i^d\| \times \|z_i(0)-\tilde{z}_i-z_i^d\|}$, $\hat{\gamma} \in [-\pi, \pi]$. For the relevant principle, see our prior work [20].

**Remark 3.** *Bidirectional range constraint (i.e., terminal return cost) is primarily considered through the return-to-base requirement in the terminal cost. This ensures that the UAV automatically meets the safe return requirement while optimizing inspection efficiency.*

**Remark 4.** *Compared to standard optimization algorithms [6,9], this approach better captures dynamic strategic interactions and ensures fairness. The outcome is a Pareto improvement where system-wide optimization is achieved without sacrificing individual utilities. The theoretical guarantees and interpretability of the NE further underscore its reliability for real-world applications such as UAV-based inspection of offshore wind turbines. As an extension of prior work [20], this study incorporates the return-trip energy constraint $\psi(z_i(t_f))$ during UAV inspections on the basis of existing literature, making it more aligned with the practical scenario of offshore wind turbine inspection.*

## The optimal coordination control strategy

Following the problem formulation in the preceding section, this section is devoted to a detailed description of the solution approach for the game-theoretic models. The L-NE for the DDG defined in Problem 1 corresponds to a set of control strategies $\{u_i^*, u_{-i}^*\}$ where each UAV $i$'s strategy $u_i^*$ is the optimal response to its neighbors' optimal strategies $u_{-i}^*$, minimizing its individual cost $J_i$ (Eq. (9)). This constitutes a coupled optimal control problem for each agent.

### Necessary conditions via Pontryagin's minimum principle

Define an auxiliary state variable as:

$$\hat{g}_i(t) \triangleq \psi(z_i(t)) + \int_0^t \left( \|Y_i(z_i)\|_{Q_i}^2 + \|u_i(t)\|_{R_i}^2 + \sum_{j \in \mathcal{N}_i} \|u_{ij}\|_{R_{ij}}^2 \right) dt. \tag{16}$$

where $\hat{g}_i(0) = \psi(z_i(0))$, $\hat{g}_i(t_f) = J_i\left(z(0), u_i^*, u_{-i}^*\right)$.

Then, we give the following form:

$$\bar{g}_i(t) = [\hat{g}_i(t), z_i^T(t)]^T \in \mathbb{R}^9. \tag{17}$$

where $\bar{g}_i(0) = [\psi(z_i(0)), z_i^T(0)]^T$.

Therefore, determining the DDG for UAV $i$ in (9) is equivalent to formulating and solving an optimal control problem, i.e.,

$$\min_{u_i} J_i\left(z(0), u_i^*, u_{-i}^*\right) = \hat{g}_i(t_f) \tag{18}$$

s.t.

$$\begin{cases} \dot{\bar{g}}_i(t) = \hat{h}(\hat{g}_i(t), \dot{z}_i(t)), \\ \bar{g}_i(0) = [\psi(z_i(0)), z_i^T(0)]^T. \end{cases}$$

where $\hat{h}(\hat{g}_i(t), \dot{z}_i(t)) = [\dot{\hat{g}}_i(t), \dot{z}_i(t)]$.

Furthermore, we determine the optimal coordinated control strategy through Pontryagin's Minimum Principle (PMP), with the corresponding Hamiltonian given by:

$$H_i(t, \lambda_i(t), z^*(t), u_i(t), u_{-i}^*(t)) \triangleq L_i + \lambda_i^T(t)\dot{z}_i, \tag{19}$$

where $L_i = \|Y_i(z_i)\|_{Q_i}^2 + \|u_i(t)\|_{R_i}^2 + \sum_{j \in \mathcal{N}_i} \|u_{ij}\|_{R_{ij}}^2$, $z^*(t)$ corresponds to the multi-UAVs' optimal state trajectory, and $\lambda_i(t) : [0, t_f] \to \mathbb{R}^8$ is the associated costate function.

Accordingly, inspired by the work in Ref. [22], Eq. (19) provides a key necessary condition for the optimal control in the DDG (9), ensuring convergence of the strategy set to an L-NE. The PMP states that for the optimal trajectory $z^*(t)$ and control $u_i^*(t)$, there exists a costate $\lambda_i(t)$ such that:

$$\begin{cases} u_i^* = \arg\min_{u_i} H_i(t, \lambda_i(t), z^*(t), u_i(t), u_{-i}^*(t)), \\ \dot{z}^* = Az + \sum_{i=1}^{N} \left( B_i u_i + \hat{\hat{g}}_i \right), \\ \dot{\lambda}_i(t) = -\frac{\partial H_i(t, \lambda_i(t), z^*(t), u_i(t), u_{-i}^*(t))}{\partial z} \end{cases} \tag{20}$$

with the boundary condition

$$\begin{cases} z^*(0) = z_0, \\ \lambda_i(t_f) = \frac{\partial \psi(z_i(t))}{\partial z_i}. \end{cases} \tag{21}$$

where $z_0$ corresponds to the system state at time $t = 0$. These conditions define a two-point boundary value problem (TPBVP) whose solution characterizes the L-NE.

### Distributed Dradient optimization for L-NE

Solving the coupled TPBVP (20) directly in a distributed manner is challenging. Inspired by the Ref. [16] for solving TPBVP, Instead, we adopt a direct optimization approach: each UAV $i$ iteratively improves its control trajectory $u_i(t)$ to minimize $J_i$ directly, using only local information. This approach is numerically robust and naturally parallelizable.

We employ a distributed gradient-based optimization algorithm, summarized in the following Algorithm 1. The control trajectory is parameterized over $[0, t_f]$. Each iteration involves: 1) Forward simulation: Integrate the dynamics (Eq. (4)) with the current control to obtain the state trajectory. 2) Gradient computation: Compute the gradient $\nabla_{u_i} J_i$ efficiently using the adjoint method. This requires a backward integration of an adjoint equation, which is computationally inexpensive and avoids explicit solution of the costate equation in (20). 3) Gradient update and communication: Update $u_i$ along the negative gradient direction, then exchange the updated control with neighbors.

**Remark 5.** *Upon convergence, the solution obtained by the gradient-based optimization algorithm satisfies the first-order necessary conditions for a minimum of the cost functional $J_i$. These conditions are mathematically equivalent to the set of Pontryagin's Minimum Principle (PMP) conditions given in Eq.(20). Specifically, the adjoint variable $\rho_i(t)$ introduced in the gradient computation obeys the same linear differential equation and terminal condition as the PMP costate $\lambda_i(t)$. Because both variables satisfy an identical linear boundary-value problem, the uniqueness theorem for such problems guarantees that $\rho_i(t) \equiv \lambda_i(t)$ at convergence. Consequently, the trajectories generated by our distributed gradient algorithm fulfill all PMP necessary conditions and therefore constitute a L-NE.*

*Even when the underlying system dynamics are nonlinear, the costate (or adjoint) equation remains a linear differential equation in $\lambda_i(t)$ (or $\rho_i(t)$). This linearity follows from the fact that the equation is derived either by linearizing the original Hamiltonian system around the optimal trajectory or directly from the variational principle. Hence, the uniqueness argument holds in the general nonlinear setting, ensuring the equivalence between the numerical solution of the gradient algorithm and the analytical PMP formulation.*

The pseudo-code of the distributed gradient optimization for L-NE is as follows.

### Algorithm 1 Distributed gradient optimization for L-NE

**Input:** Initial state $z_i(0)$, neighbor strategies $u_{-i}^*$, cost weights, horizon $t_f$, step size $\eta_0$, tolerance $\varepsilon$.
**Output:** Optimal control $u_i^*$.
1: **Repeat**

2: **Forward simulation** Integrate the dynamics $\dot{z}_i = a_i z_i + b_{ii} u_i^{(k)} + \hat{g}_{ii}$ (Eq. 4) forward from $t=0$ to $t_f$, obtaining the state trajectory $z_i^{(k)}(t)$.
3: **Gradient computation** Compute the cost gradients $\frac{\partial L_i}{\partial z_i}$ and $\frac{\partial L_i}{\partial u_i}$ along $z_i^{(k)}(t)$.
4: Integrate the adjoint equation backward in time: $-\dot{\rho}_i = a_i^T \rho_i + \frac{\partial L_i}{\partial z_i}$ with terminal condition $\rho_i(t_f) = \frac{\partial \psi}{\partial z_i(t_f)}$.
5: Compute the gradient: $\nabla_{\mathbf{u}_i} J_i^{(k)} = \int_0^{t_f} \left( b_{ii}^T \rho_i(t) + \frac{\partial L_i}{\partial u_i} \right) dt$.
6: **Distributed communication** Broadcast the updated control $\mathbf{u}_i^{(k)}$ to all neighbors $j \in \mathcal{N}_i$.
7: Receive neighbors' controls $\mathbf{u}_j^{(k)}$ for $j \in \mathcal{N}_i$.
8: **Projected gradient update**
9: $\mathbf{u}_i^{(k+1)} \leftarrow \mathcal{P}_{\mathcal{U}}\left[ \mathbf{u}_i^{(k)} - \eta_k \cdot \nabla_{\mathbf{u}_i} J_i^{(k)} \right]$, where $\mathcal{P}_{\mathcal{U}}$ projects onto the feasible control set.
10: Update step size $\eta_k$ via backtracking line search.
11: $k \leftarrow k+1$.
12: Until $\|\nabla_{\mathbf{u}_i} J_i^{(k)}\| < \epsilon$ for all $i$ **or** $k > K_{\max}$.

The L-NE strategy $u_i^*$ is obtained by numerically minimizing the cost function $J_i$ in (9) subject to the dynamics constraint (4). This is achieved using a distributed gradient descent algorithm, which directly optimizes the control trajectory without explicitly solving the two-point boundary value problem for the costate $\lambda_i(t)$.

The Pontryagin's Minimum Principle (PMP) applied to our DDG formulation yields the set of necessary conditions for optimality (Eqs. 20–21). These conditions, which include a two-point boundary value problem, define what constitutes a Local Nash Equilibrium (L-NE). The theoretical contribution of our work (Proposition 1) is to prove that under a strongly connected graph, the unique solution satisfying these local conditions for all agents converges to a Global Nash Equilibrium (G-NE).

**Remark 6.** *To illustrate the scalability of the proposed DDG method, an analysis is conducted from two aspects: computational burden and real-time feasibility.*

1) *Computational burden: Solving the local optimal control problem (Eq. 18–20)for each UAV involves a two-point boundary value problem with state dimension 8. We employ an efficient iterative solver (i.e., a gradient-based method) whose convergence per agent typically requires $\mathcal{O}(10^2)$ iterations in our simulations, with each iteration involving low-dimensional matrix operations [16]. The distributed architecture allows these computations to be parallelized across UAVs.*

2) *Real-time feasibility: For the inspection scenarios considered (mission duration 400s), the trajectory planning is computed offline or re-planned at low frequency (every 30s) based on updated neighbor states. The per-agent computation time (0.5s on a standard desktop CPU) is negligible compared to the re-planning interval, demonstrating the method's potential for near real-time operation.*

We also compare our proposed approach with a centralized game-theoretic solver, which employs the same PMP principle and cost structure as our DDG method but solves a single, high-dimensional optimization problem using global information (i.e., centralized DG). This centralized solver must handle the concatenated state vector of all N UAVs, resulting in a total state dimension of 8N. Consequently, its computational complexity scales approximately as $\mathcal{O}((N \times n)^3)$, where $n = 8$ represents the state dimension of a single UAV.

## The G-NE

The primary objective of the multi-UAV inspection mission for offshore wind farms is to minimize the total inspection time. This goal necessitates globally optimal coordination of the entire fleet, surpassing what individual UAVs can achieve locally. Consequently, the control strategy must ensure that the system converges to a G-NE, which guarantees the time-optimal performance for the entire mission, as supported by Ref. [22]. The following definition of G-NE is formalized to this end.

**Definition 1.** *(G-NE) An N-tuple of coordination strategies $\{u_1^*, \cdots, u_N^*\}$ for the N-UAV inspection game constitutes a G-NE if, for every UAV i, the following conditions are met:*

1) *Optimality condition: The control strategy $u_i^*$ is the optimal response to the optimal strategies $u_i^*$ of all other UAVs:*

$$J_i\left(z(0), u_i^*, u_i^*\right) \leq J_i\left(z(0), u_i, u_i^*\right), \quad \forall u_i \neq u_i^*.$$
(22)

2) *Non-Triviality condition: There exists an alternative strategy $\breve{u}_i$ such that a unilateral deviation from $u_i^*$ results in a different system cost:*

$$J_i\left(z(0), u_i^*, u_i^*\right) \neq J_i\left(z(0), \breve{u}_i, u_i^*\right), \quad \text{for } \breve{u}_i \neq u_i^*.$$
(23)

Next, the following proposition concerning the convergence of a L-NE to a G-NE is presented.

**Proposition 1.** *(Convergence of L-NE to G-NE) Under a strongly connected communication topology $G(\mathcal{V}, \varepsilon)$, let $u_i^*(\forall i \in \mathbb{N}_{1:N})$ denote the optimal coordinated control strategy of UAV $i$, derived from its interactions with neighbors $j(j \in \mathcal{N}_i)$. If the distributed gradient algorithm (Algorithm 1) converges, and the communication graph $G(\mathcal{V}, \varepsilon)$ is strongly connected, then the L-NE generated by the algorithm will converge to a G-NE, i.e.,*

$$J_i\left(z(0), u_i^*, u_{-i}^*\right) = J_i\left(z(0), u_i^*, u_i^*\right), \quad \forall i \in \mathbb{N}_{1:N}.$$
(24)

**Proof.** The proof is divided into three steps: (1) the algorithm converges to an L-NE (satisfying the PMP); (2) strong connectivity enforces global consistency through gradient exchange; (3) convexity ensures that the local solution is unique, and thus globally unique.

Step 1 (Algorithm convergence and attainment of the L-NE): For each UAV $i$, integrate the dynamics (Eq. (4)) over $[0, t_f]$ using the current control strategy $u_i^{(k)}(t)$.

$$\begin{cases} \dot{z}_i^{(k)}(t) = a_i z_i^{(k)}(t) + b_{ii} u_i^{(k)}(t) + \hat{\bar{g}}_{ii}, \\ z_i^{(k)}(0) = z_{i0}. \end{cases}$$
(25)

To efficiently compute the gradient $\nabla_{u_i} J_i^{(k)}$, an adjoint variable $\lambda_i(t) \equiv \rho_i(t) \in \mathbb{R}^8$ is introduced, governed by the adjoint equation (which provides an efficient computation of the costate equation in (20)):

$$\begin{cases} -\dot{\lambda}_i^{(k)}(t) = a_i^T \lambda_i^{(k)}(t) + Q_i \frac{\partial}{\partial z}\left(\|Y_i(z_i)\|^2\right), \\ \lambda_i^{(k)}(t_f) = F_i\left(z_i^{(k)}(t_f) - z_{ib}\right). \end{cases}$$
(26)

After backward integration, the gradient is obtained from the partial derivative of the Hamiltonian:

$$\nabla_{u_i} J_i^{(k)}(t) = R_i u_i^{(k)}(t) + b_{ii}^T \lambda_i^{(k)}(t).$$
(27)

Each agent updates its strategy along the negative gradient direction:

$$u_i^{(k+1)}(t) = u_i^{(k)}(t) - \eta \cdot \nabla_{u_i} J_i^{(k)}(t).$$
(28)

where $\eta > 0$ is the step size. Updated strategies $u_i^{(k+1)}$ are then broadcast to all neighbors $j(j \in \mathcal{N}_i)$. Upon convergence, for all $i \in \mathbb{N}_{1:N}$,

$$\nabla_{u_i} J_i^{(\infty)}(t) = R_i u_i^*(t) + b_{ii}^T \lambda_i^*(t) = 0, \quad \forall t \in [0, t_f].$$
(29)

According to PMP, this condition, together with the state equation, adjoint equation, and transversality condition, constitutes the first-order necessary condition for optimality. In the distributed setting, this implies that for each agent $i$, given the optimal strategies of its neighbors $u_{-i}^*$, the strategy $u_i^*$ is a local minimizer of its individual cost (9), thereby satisfying the definition of a L-NE (8).

Step 2 (Global consistency enforced by strong connectivity): Assume the communication graph $G(\mathcal{V}, \varepsilon)$ is strongly connected. Suppose, for contradiction, that the L-NE strategies are not globally consistent.(i.e., there exist two disjoint subsets of UAVs whose locally optimal solutions are mutually incompatible given the global mission objectives.) Such inconsistency would manifest as a mismatch in the coupled cost terms via the $R_{ij}$ terms in (9). For any adjacent UAVs $i$ and $j(j \in \mathcal{N}_i)$, a strategy discrepancy would produce a non-zero gradient component:

$$\left\| \frac{\partial}{\partial u_i} \sum_{j \in \mathcal{N}_i} \|u_{ij}\|_{R_{ij}} \right\| \neq 0.$$
(30)

During the iterative process, this gradient information is exchanged among neighbors (Step 3 of Algorithm 1). Strong connectivity guarantees that there exists a directed path from any agent $i$ to any other agent $l$. Consequently, any local inconsistency (nonzero gradient) propagates through the entire network via successive neighbor-to-neighbor exchanges.

Define the global gradient norm as $\Phi^{(k)} = \sum_{i=1}^{N} \int_0^{t_f} \left\| \nabla_{u_i} J_i^{(k)}(t) \right\|^2 dt$. The gradient-descent update ensures that $\Phi^{(k)}$ is non-increasing with $k$. At convergence,

$$\lim_{k \to \infty} \Phi^{(k)} = 0.$$
(31)

Then,

$$\nabla_{u_i} J_i^{(\infty)}(t) = 0.$$
(32)

This global zero-gradient condition implies that not only each agent's own gradient vanishes, but also all coupled interaction terms (via $R_{ij}$) are balanced, thereby eliminating any pairwise strategic contradictions. Hence, the locally optimal strategies $u_i^*$ are globally consistent.

Step 3 (Uniqueness and attainment of the G-NE): To establish the uniqueness of the G-NE, we begin by analyzing the adjoint system derived from a quadratic approximation of the problem around the equilibrium trajectory. Consider the Nash equilibrium trajectory $z^*$, Defining the deviation as $\hat{z} = z - z^*$, we have the collective state vector $\hat{z} = [\hat{z}_1^T, \cdots, \hat{z}_N^T]^T \in \mathbb{R}^{8N}$ and the co-state vector $\lambda = [\lambda_1^T, \cdots, \lambda_N^T]^T \in \mathbb{R}^{8N}$. The associated adjoint system with two-point boundary values is given by (The derivation is detailed in S1 Appendix):

$$\begin{bmatrix} \dot{\hat{z}}(t) \\ \dot{\lambda}(t) \end{bmatrix} = \begin{bmatrix} 0 & -I_{8N} \\ -\hat{\Omega} & 0 \end{bmatrix} \begin{bmatrix} \hat{z}(t) \\ \lambda(t) \end{bmatrix},$$

$$\begin{bmatrix} \hat{z}(0) \\ \lambda(t_f) \end{bmatrix} = \begin{bmatrix} \hat{z}(0) \\ Fz(t_f) \end{bmatrix},$$
(33)

where the matrix $\hat{\Omega} \in \mathbb{R}^{8N \times 8N}$ is a block-diagonal matrix, $\hat{\Omega} = diag\{\hat{\Omega}_1, \cdots, \hat{\Omega}_i, \cdots, \hat{\Omega}_N\}$. Each block is constructed from the Hessian of the running cost function for UAV $i$ evaluated at the equilibrium:

$$\hat{\Omega} = \sum_{i=1}^{N} \alpha_i \nabla^2 Y_i(z_i^*).$$

<div align="right">(34)</div>

where $z_i^*$ is the state of UAV $i$ at the Nash equilibrium, $\nabla^2 Y_i(z_i^*)$ is the Hessian matrix of $Y_i$ at $z_i^*$, $\alpha_i > 0$ is a positive weighting coefficient. The aggregate matrix $\hat{\Omega}$ thus represents a weighted sum of the individual Hessians.

The positive definiteness of $\hat{\Omega}$ is crucial and follows from the construction of the running cost $Y_i(z_i)$. This cost combines a collision avoidance penalty $D(z_i)$ and a trajectory optimization term $DS(z_i(t))$. The penalty term $D(z_i)$ is designed to be convex and increasing outside a safe distance, with a positive definite Hessian at the collision-free equilibrium point $z_i^*$. The trajectory term $DS(z_i(t))$ is also convex (e.g., based on squared angular deviation), yielding a positive semi-definite Hessian. By selecting positive coefficients $\alpha_i$, the weighted superposition of these terms ensures that each $\nabla^2 Y_i(z_i^*)$ is positive semi-definite, with at least one block being positive definite. Consequently, the block-diagonal matrix $\hat{\Omega}$ is positive definite.

The positive definiteness of $\hat{\Omega}$ together with the positive definiteness of the weighting matrices $Q_i, R_i, R_{ij}, F_i$ in the cost function (9), guarantees that the integrated cost term $\|Y_i(z_i)\|_{Q_i}^2 + \|u_i(t)\|_{R_i}^2 + \sum_{j \in \mathcal{N}_i} \|u_{ij}\|_{R_{ij}}^2$ and the terminal cost $\psi(z_i(t))$ are jointly convex. Therefore, given the strategies of its neighbors $u_{-i}^*$, each UAV's optimization problem is strictly convex. For a strictly convex problem, any point satisfying the first-order necessary condition (i.e., the zero-gradient condition) is the unique global minimizer. Thus, the strategy $u_i^*$ obtained in Step 1 is the unique optimal response of UAV $i$ to $u_{-i}^*$, each UAV's optimization subproblem is strictly convex. For a strictly convex problem, any point satisfying the first-order necessary optimality conditions (i.e., the zero-gradient condition derived from Pontryagin's Minimum Principle) is the unique global minimizer. Hence, the strategy $u_i^*$ obtained upon convergence of the distributed gradient algorithm is the unique optimal response of UAV $i$ to its neighbors' strategies $u_i^*$.

As established in Step 2 of the main proof, the strong connectivity of the communication graph $G(\mathcal{V}, \varepsilon)$, ensures that the locally optimal strategies are globally consistent. The collection of these unique local optimal responses, $u^* = (u_1^* \cdots u_i^* \cdots u_N^*)$, therefore forms a strategy profile that satisfies the definition of a G-NE (Definition 1). Formally, for every UAV $i$, if all other UAVs adhere to $u_i^*$ (the strategies of all agents except $i$), then $u_i^*$ is its optimal response:

$$J_i(z(0), u_i^*, u_i^*) \leq J_i(z(0), u_i, u_i^*), \quad u_i^* \neq u_i.$$

<div align="right">(35)</div>

The distinction between the L-NE and the G-NE is that the G-NE considers the strategies of all other UAVs, not just immediate neighbors. The strong connectivity of the network, which enables the propagation of local consistency, guarantees the equivalence between these two notions in our framework. Consequently, the uniqueness of the solution to the adjoint system (ensured by $\hat{\Omega} > 0$), combined with the strong connectivity of $G(\mathcal{V}, \varepsilon)$, secures the convergence of the algorithm to a G-NE:

$$J_i\left(z(0), u_i^*, u_{-i}^*\right) = J_i(z(0), u_i^*, u_i^*), \quad \forall i \in \mathbb{N}1 : N.$$

<div align="right">(36)</div>

In summary, under a strongly connected communication topology, the distributed gradient algorithm converges to a strategy profile that is both a local and a global Nash Equilibrium. □

**Remark 7.** *The strong connectivity of the communication graph $G(\mathcal{V}, \varepsilon)$ ensures a bidirectional information path between any two UAVs, allowing local strategy information to propagate across the entire network in finite time. This leads to global alignment of the L-NE strategies and consequently drives convergence toward a G-NE. Furthermore, the positive definiteness of $\hat{\Omega}$ guarantees the existence and uniqueness of the L-NE, which in turn ensures that the resulting G-NE is also uniquely defined.*

## Simulations

Here we conduct a comprehensive assessment of the proposed DDG method's core capabilities: coordinated optimality and operational safety. A systematic comparison against two benchmark methods is presented to demonstrate how

coordinated behavior enhances mission efficiency through reduced completion times without compromising safety in maritime multi-UAV inspection scenarios: The GA-DZ method, which optimizes UAV trajectories by minimizing the total flight distance, exhibits limited adaptability and fails to account for bidirectional range constraints [6], and the NN-DRL method, which is prone to convergence to local optima [9].

**Remark 8.** *This study focuses specifically on the operational control of multi-UAV systems during maritime inspection, and thus the simulations are confined to the operational space in which UAVs execute inspection tasks over offshore wind turbines. Given that target wind turbines are pre-assigned to each UAV, the core objective is to ensure the safe and efficient completion of these missions, rather than addressing the task allocation problem. To guarantee a fair comparison, all evaluated methods—including the proposed approach and the benchmarks—utilize the same initial task assignments and are implemented with fully disclosed parameters.*

For the GA-DZ baseline, we adopted the genetic algorithm with dynamic zoning as outlined in Ref. [6], utilizing the authors' publicly available source code. The algorithm encodes solutions as sequences of waypoint assignments and corresponding flight trajectories. Its optimization is driven by a fitness function defined as the inverse of the total path distance for the UAV fleet. To ensure a meaningful comparison that tests the algorithm's inherent ability to satisfy constraints, substantial penalties ($1 \times 10^6$ per violation) are applied in the fitness evaluation for any path that violates safety distances limits. The evolutionary process uses a population size of 100 and runs for 500 generations per simulation, with a crossover rate of 0.85, a mutation rate of 0.1, and tournament selection (size=3). Unlike learning-based methods, GA-DZ does not involve a separate training phase; it is executed directly on each test scenario to produce a scenario-specific solution, enabling a direct and fair performance comparison under the same conditions as the proposed DDG method.

For the NN-DRL baseline, we implemented a Dueling Deep Q-Network (DQN) following the architecture in Ref. [9]. The network takes the UAV's state as input, processes it through two fully-connected layers, and outputs Q-values for each discrete action. The state space includes the UAV's pose $p_i$, velocity $v_i$, remaining battery, relative positions to its target and the nearest wind turbine, as well as positions of neighboring UAVs within communication range. The action space is defined as nine discrete actions: hovering and moving in eight fixed-speed directions. The reward function is designed to balance multiple objectives and is formulated as:

$$\hat{r}_i = -\varkappa_1 d_i - \varkappa_2 \bar{e}_i + \varkappa_3 \hat{e}_i - \varkappa_4 \hat{c}_i - \varkappa_5 \hat{f}_i, \quad \forall i \in \mathbb{N}_{1:N}. \tag{37}$$

where $d_i$, $\bar{e}_i$, $\hat{e}_i$, $\hat{c}_i$ and $\hat{f}_i$ denote distance to target, energy consumption, inspection completion reward, collision penalty, and return violation penalty, respectively, with all coefficients tuned accordingly. During training, an $\varepsilon$-greedy exploration strategy is adopted, with $\varepsilon$ linearly decaying from $1.0$ to $0.05$ over the first $8,000$ episodes. Each agent was trained for $10,000$ episodes on a randomized set of training scenarios to ensure strategy generalization. The trained strategy was then evaluated on a separate, held-out test set that is identical to the scenarios used for evaluating the proposed DDG method, thereby ensuring a fair comparison. The parameters of the NN-DRL method are summarized in Table 2.

**Table 2. The parameters of the NN-DRL method [9].**

| Parameter | Value |
|---|---|
| Learning rate | 0.001 |
| Discount factor | 0.99 |
| Replay buffer size | 50000 |
| Batch size | 64 |
| Target update frequency | 100*step* |
| Total number of training episodes | 10000 |

We consider a multi-UAV inspection system comprising three UAVs (i.e., $N = 3$). The environment contains 28 offshore wind turbines and one offshore booster station (OBS), which serves as both the charging base and the common starting point for all UAVs. The simulated obstacles include known wind turbine towers, each with a radius of *5m*, as well as other UAVs, which are treated as unknown dynamic obstacles. Each UAV is assigned a specific subset of wind turbines for inspection and plans its trajectory accordingly. To ensure a fair comparison, the proposed DDG method and the two benchmark approaches [6,9] are evaluated under identical conditions: the same initial positions and dynamics model (Eq. 3), the same environmental layout (28 turbines and 1 OBS), and the same success criteria (complete inspection, collision-free operation, and safe return to the OBS). This setup isolates the performance differences to the algorithmic level. All other relevant simulation parameters of the proposed DDG are summarized in Table 3.

As shown in Fig 3, which shows the inspection planned trajectories under three methods. UAV1 is responsible for inspecting wind turbine set $\{1, 4, 7, 10, 13, 16, 19, 22, 25, 28\}$; UAV2 for set $\{2, 5, 8, 11, 14, 17, 20, 23, 26\}$; and UAV3 for set $\{3, 6, 9, 12, 15, 18, 21, 24, 27\}$. For clarity, the inspection sequences with the three methods are summarized in a Table 4. The results indicate that the proposed method achieves the shortest total path length of 845 m. This is because the proposed method transforms the estimated planning of the UAVs into an optimally coordinated DDG model, obtaining a G-NE trajectory that balances maximum flight range and reduced inspection time. The GA-DZ method [6], which uses a genetic algorithm to optimize trajectories and aims to reduce inspection time, neglects the safety constraints of the UAVs. This makes it unsuitable for densely distributed wind turbine scenarios. During inspection, the wind turbines remain operational. When UAV1 using the GA-DZ method flies from turbine 1 to turbine 4, it collides with turbine 4 due to unaccounted dynamic obstacles, preventing completion of subsequent tasks

**Table 3. The other related simulation parameters of the proposed DDG.**

| Parameter | Value |
|---|---|
| $m_i$ | $1.5kg$ |
| $I_{ixx}$ | $0.03kg \cdot m^2$ |
| $I_{iyy}$ | $0.03kg \cdot m^2$ |
| $I_{izz}$ | $0.06kg \cdot m^2$ |
| $L_i$ | $0.15m$ |
| $\hat{b}_i$ | $1.5 \times 10^{-5}N \cdot s^2$ |
| $\hat{d}_i$ | $2.5 \times 10^{-7}N \cdot m \cdot s^2$ |
| $g_i$ | $9.81m/s^2$ |
| $\hat{R}_i$ | $4m$ |
| $v_{imax}$ | $10m/s$ |
| $u_{imin}$ | $[0, -1, -1, -1]^T N$ |
| $u_{imax}$ | $[30, 1, 1, 1]^T N$ |
| $v_i(0)$ | $[0, 2, 0, 0]^T m/s$ |
| $Q_i$ | $0.1 \times I_8$ |
| $R_i$ | $I_4$ |
| $R_{ij}$ | $0.05I_4$ |
| $F_i$ | $10I_8$ |
| $\hat{\alpha}_1$ | $0.3$ |
| $\hat{\alpha}_2$ | $0.1$ |
| $\bar{R}_i$ | $20m$ |
| Stopping criterion | $\|\tilde{z}_i\| \leq 0.01$ |

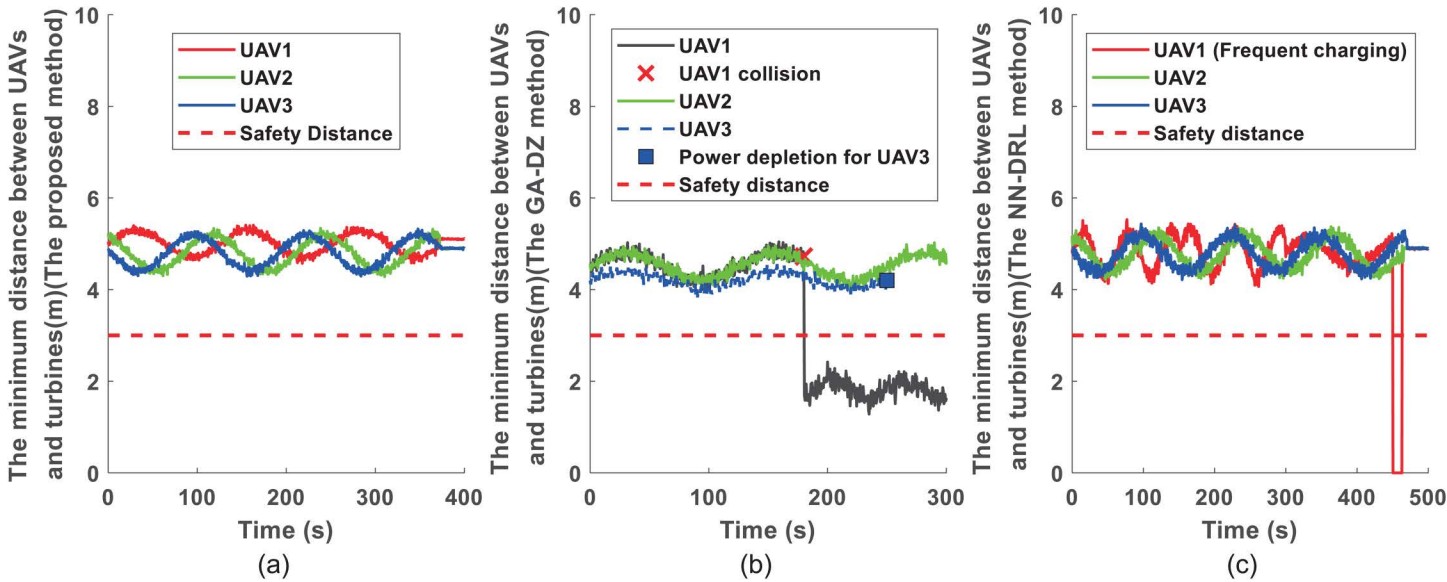

Fig 3. 3 UAVs and 28 turbines. (a) The minimum distance between UAVs and wind turbines with the proposed method; (b) The minimum distance between UAVs and wind turbines with the GA-DZ method; (c) The minimum distance between UAVs and wind turbines with the NN-DRL method.

Table 4. The inspection sequences and path length with the three methods.

| Method | Inspection sequences |
|---|---|
| The proposed method | UAV1:{28 → 4 → 1 → 19 → 22 → 7 → 25 → 10 → 16 → 13}(311$m$) |
| | UAV2: {23 → 20 → 5 → 2 → 26 → 17 → 14 → 11 → 8}(278$m$) |
| | UAV3: {18 → 6 → 12 → 24 → 15 → 27 → 9 → 3 → 21} (256$m$) |
| The GA-DZ method | UAV1: {1 → 4}(Collision)(102$mm$) |
| | UAV2: {2 → 26 → 23 → 20 → 17 → 14 → 11 → 8 → 5}(325$m$) |
| | UAV3: {3 → 6 → 9 → 12} (Power depletion)(136$m$) |
| The NN-DRL method | UAV1:<br>{13 → 10 → 7 → 4 → 1 → *Charging* → 16 → 19 → 22 → 25 → *Charging* → 28}<br>(457$m$) |
| | UAV2: {2 → 26 → 5 → 8 → 17 → 14 → 17 → 20 → 23}(316$m$) |
| | UAV3: {27 → 24 → 21 → 18 → 15 → 12 → 9 → 6 → 3}(294$m$) |

(Fig 3(b), red solid line). Additionally, UAV3 fails to complete its inspection because it runs out of battery while flying from turbine 9 to turbine 12, as the return energy constraint is not considered (Fig 3(b), blue dashed line). The NN-DRL method [9] yields the longest total path length of 10679 m. This is attributed to its tendency to fall into local optima during online trajectory planning. Although it considers the return energy constraint, frequent returns for recharging reduce inspection efficiency. For instance, UAV1 returns to charge three times, significantly increasing the total path length (Fig 3(c), red solid line).

To evaluate safety during inspection, Fig 4 shows the minimum distances between each UAV and obstacles (wind turbines and other UAVs within the field of view) for the three methods. Fig 4(a) corresponds to the proposed method, where all UAVs maintain distances greater than the safe threshold. Fig 4(b) illustrates the results for the GA-DZ method [6]. While UAV2 maintains safe distances, the minimum distance between UAV1 and wind turbine 4 falls below the safe

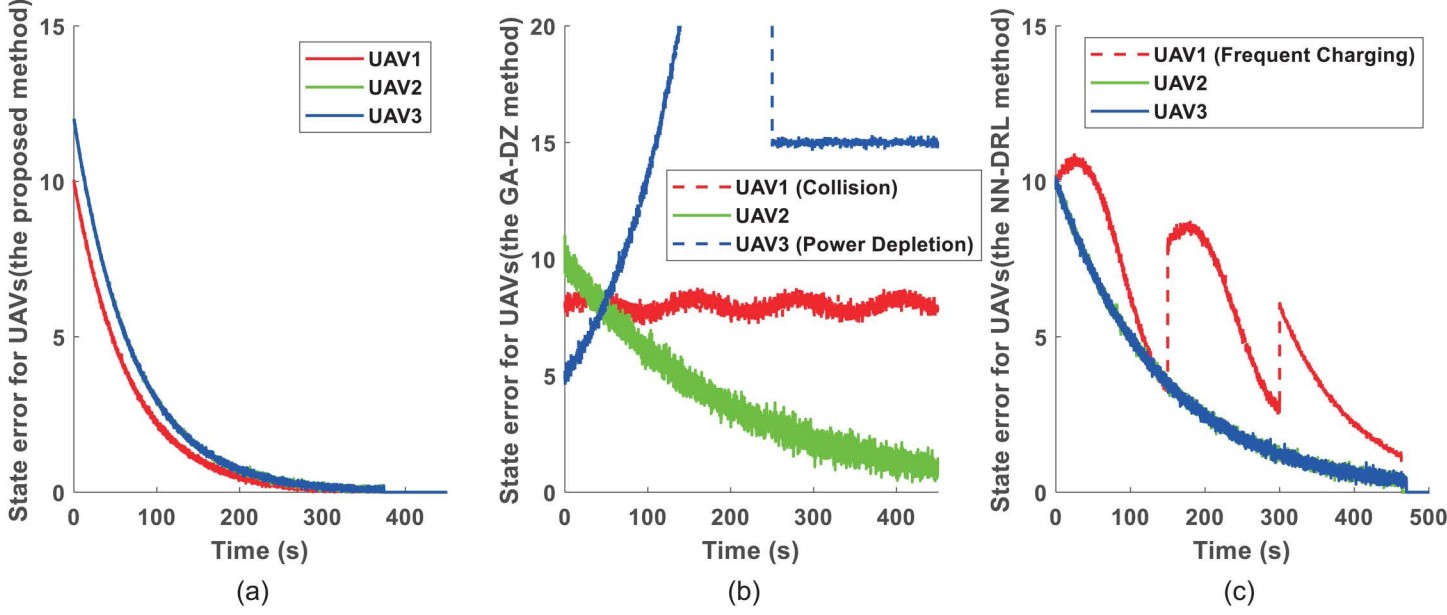

**Fig 4. The state errors for 3 UAVs and 28 turbines. (a)** under proposed approach; **(b)** under the GA-DZ approach; **(c)** under the NN-DRL approach.

threshold, indicating a collision. Moreover, UAV3 stops operating at 250 s due to battery depletion. Fig 4(c) presents the results for the NN-DRL method [9], where all UAVs maintain distances above the minimum safe level.

Fig 5 further compares the state error convergence of the three methods. Fig 5(a) shows the evolution of state errors for each UAV using the proposed approach. All errors converge to zero, confirming successful completion of the inspection tasks. Fig 5(b) displays the state errors for the GA-DZ method [6], where only UAV2 completes its task. Fig 5(c) presents the state errors for the NN-DRL method [9], where all UAVs' errors converge to zero, indicating task completion.

To demonstrate the scalability and effectiveness of the proposed method, a scenario with six UAVs inspecting 40 wind turbines is designed. UAV1 is assigned turbines {1, 7, 13, 19, 25, 31, 37}; UAV2: {2, 8, 14, 20, 26, 32, 38}; UAV3: {3, 9, 15, 21, 27, 33, 39}; UAV4: {4, 10, 16, 22, 28, 34, 40}; UAV5:{5, 11, 17, 23, 29, 35}; and UAV6: {6, 12, 18, 24, 30, 36}. Fig 6 shows the inspection trajectories for the three methods, with sequences summarized in a Table 5. The proposed method again achieves the shortest total path length (272 + 226 + 206 + 312 + 240 + 308 = 1564 m). UAV1 returns to recharge when its battery drops below a threshold and then resumes inspection. With the GA-DZ method [6], UAV1 collides with turbine 16 while flying from turbine 7 to the next target, preventing further inspection (Fig 6(b), red solid line). UAV3 fails to complete its task due to insufficient battery (Fig 6(b), blue dashed line). The NN-DRL method [9] produces the longest total path length, with UAV1 and UAV4 each returning to charge twice, significantly increasing the path length (Fig 6(c), red and yellow solid lines).

Similarly, Fig 7 plots the minimum distances between each UAV and obstacles. Fig 7(a) shows that all UAVs maintain safe distances with the proposed method. Fig 7(b) illustrates that for the GA-DZ method [6], UAV1 violates the safe distance, and UAV3 stops at 180 s due to battery depletion. Fig 7(c) shows that all UAVs maintain safe distances with the NN-DRL method [9].

Fig 8 compares the state error convergence for the six-UAV scenario. Fig 8(a) shows that all UAVs' state errors converge to zero with the proposed method, indicating successful task completion. Fig 8(b) reveals that for the GA-DZ method [6], the state errors of UAV1 and UAV3 do not converge to zero, meaning these UAVs fail to complete their

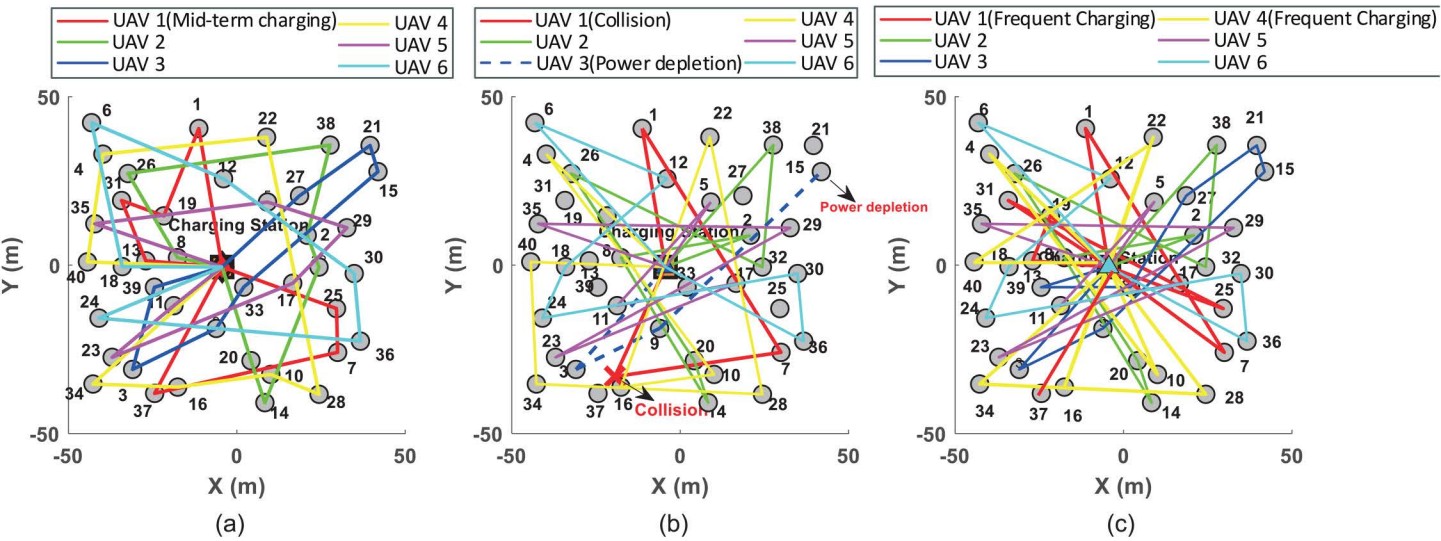

**Fig 5. Comparison of inspection trajectories for 6 UAVs and 40 turbines. (a)** Proposed DDG method: UAV1's elongated path requires it to return to the OBS mid-mission. **(b)** GA-DZ method [6]: UAV1 collides with Wind Turbine 16 (marked by 'X'), UAV3 exhausts its battery before returning. **(c)** NN-DRL method [9]: UAV1 and UAV4 exhibits an inefficient, elongated path requiring mid-mission returns to the OBS.

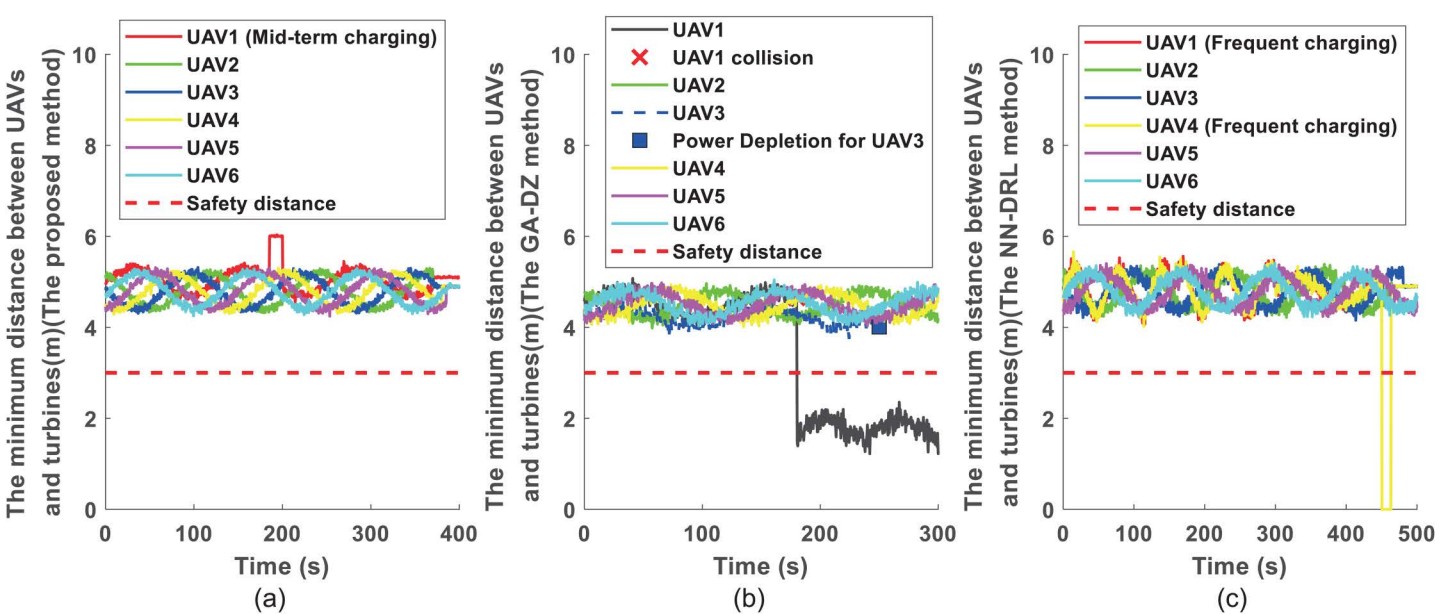

**Fig 6. 6 UAVs and 40 turbines. (a)** The minimum distance between UAVs and wind turbines with the proposed method; **(b)** The minimum distance between UAVs and wind turbines with the GA-DZ method; **(c)** The minimum distance between UAVs and wind turbines with the NN-DRL method.

inspections. Fig 8(c) shows that all UAVs' state errors converge to zero with the NN-DRL method [9], confirming task completion.

In contrast to GA-DZ's failure in safety and return energy, and NN-DRL's suboptimal time efficiency due to lack of global coordination guarantees, our method ensures both safety and time-optimality by construction through the proven G-NE.

**Table 5. The inspection sequences and path length with the three methods.**

| Method | Inspection sequences |
|---|---|
| The proposed method | UAV1:{13 → 31 → 19 → 1 → *Charging* → 25 → 7 → 37}(272*m*) |
| | UAV2: {8 → 26 → 38 → 2 → 32 → 14 → 20}(226*m*) |
| | UAV3: {39 → 3 → 9 → 33 → 15 → 21 → 27} (206*m*) |
| | UAV4: {40 → 4 → 22 → 28 → 10 → 16 → 34} (312*m*) |
| | UAV5: {11 → 23 → 17 → 29 → 5 → 35} (240*m*) |
| | UAV6: {18 → 6 → 12 → 30 → 36 → 24} (308*m*) |
| The GA-DZ method | UAV1: {1 → 7}(Collision)(208*m*) |
| | UAV2: {38 → 32 → 26 → 20 → 14 → 8 → 2}(235*m*) |
| | UAV3: {3 → 9 → 15} (Power depletion)(135*m*) |
| | UAV4: {22 → 28 → 16 → 34 → 40 → 4 → 10} (450*m*) |
| | UAV5: {35 → 29 → 23 → 17 → 11 → 5}(353*m*) |
| | UAV6: {6 → 12 → 18 → 24 → 30 → 36} (323*m*) |
| The NN-DRL method | UAV1: {1 → 7 → *Charging* → 31 → 25 → 13 → 19 → *Charging* → 37} (366*m*) |
| | UAV2: {38 → 32 → 26 → 20 → 14 → 8 → 2}(235*m*) |
| | UAV3: {39 → 33 → 27 → 21 → 15 → 9 → 3} (248*m*) |
| | UAV4: {28 → 16 → 34 → *Charging* → 4 → 10 → 40 → *Charging* → 22} (398*m*) |
| | UAV5: {35 → 29 → 23 → 17 → 11 → 5}(353*m*) |
| | UAV6: {6 → 12 → 18 → 24 → 30 → 36}(323*m*) |

To further illustrate the computational cost of the proposed method, the comparison results of online average computation time for the three methods under different paradigms are shown in Fig 9. In the scenario of 3 UAVs inspecting 28 wind turbines, the online average computation time of the proposed DDG method is 0.15 seconds, which outperforms GA-DZ [6] at 0.50 seconds and NN-DRL [9] at 0.30 seconds. The GA-DZ method suffers from significant computational burden due to its reliance on online genetic algorithm optimization. The NN-DRL method, while better than GA-DZ, still requires periodic online learning updates, resulting in higher computation times. When scaling up to 6 UAVs inspecting 40 wind turbines, the computation time of the DDG method only increases to 0.17 seconds, a growth rate of 13.3%, significantly lower than the 70.0% of GA-DZ and 50.0% of NN-DRL. This notable advantage stems from the core design of the distributed differential game framework: each UAV only needs to solve a local optimal control problem (Eq. 18–20), whose computational complexity depends solely on its own state and the number of neighbors $\mathcal{N}_i$, independent of the total system scale $N$, thereby ensuring the scalability of individual computations. At the same time, the distributed nature of the algorithm leads to a near-linear increase in the total system-wide computational load with respect to $N$, avoiding the combinatorial explosion or polynomial complexity often encountered in centralized global optimizers; the communication overhead scales with the network density (number of edges in graph $G(\mathcal{V}, \varepsilon)$), and under the assumption of a strongly connected graph, the system is guaranteed to converge to a G-NE regardless of how $N$ increases. By incorporating a local information interaction mechanism, the proposed method achieves a gradual increase in computational burden with scale, making it more suitable for real-world offshore wind farm inspection scenarios characterized by limited communication and variable scales.

To further illustrate the efficacy in minimizing inspection time without compromising safety of the proposed DDG method, we conducted a comparative validation of the three methods in 30 randomly generated simulation scenarios. In each trial, the initial positions of the UAVs and the assignment sequence of wind turbines were varied within a defined operational range. Each scenario involves 3 UAVs and 28 wind turbines, with the inspection tasks per UAV and algorithm

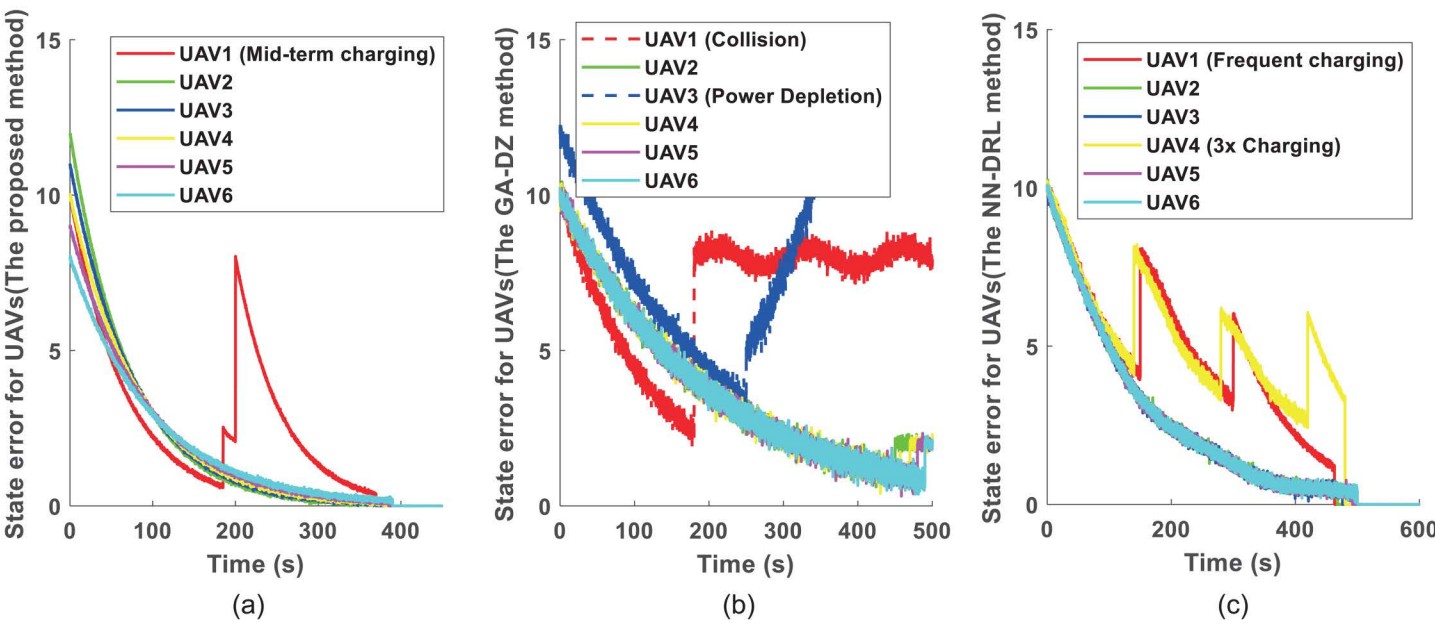

**Fig 7. The state errors for 6 UAVs and 40 turbines.** (a) under proposed approach; (b) under the GA-DZ approach; (c) under the NN-DRL approach.

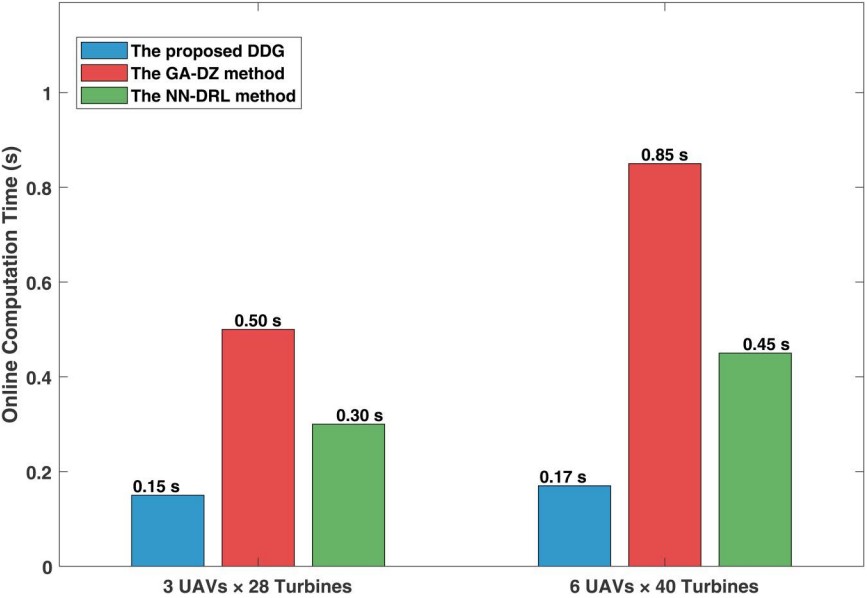

**Fig 8. Comparison of the online average computation time for the three methods under two scales.**

parameters remaining unchanged (see Table 2 to Table 3). Fig 10 visually presents the data distribution of the three methods in terms of minimum distance and task completion time. The figure includes the median (line inside the box), inter-quartile range (box range), whiskers (normal data range), and outliers (individual points), providing complete statistical distribution information.

**Fig 9. Performance distribution copmarison of three methods. (a)** The minimum distance between UAVs and wind turbines with the three methods **(b)** The task completion time between UAVs and wind turbines with the three methods.

By observing Fig 10(a), the minimum distance distributions of the proposed DDG method (median: 5.25 m, interquartile range: 4.64–5.89 m) and the NN-DRL method (median: 5.13 m, interquartile range: 4.58–5.69 m) are highly overlapping, and both are significantly above the 4.0 m safety distance (indicated by the dashed line). Both methods maintain a 100% success rate across the 30 scenarios. The box height of the DDG method is slightly narrower than that of the NN-DRL method, indicating better inter-scenario consistency in maintaining safe distance. All data points are above 4.5$m$, with no outliers. The GA-DZ method succeeded in only 6 scenarios (20% success rate), and although its minimum safe distance distribution (median: 4.13 $m$) is above the threshold, it contains multiple data points close to the lower limit. More importantly, 80% of the scenarios failed due to collisions or energy depletion, confirming the high risk of this method in practical applications.

By observing Fig 10(b), the task completion time distribution of the proposed DDG method (median: 420.0 s, interquartile range: 357.9–561.0 s) is entirely lower than that of the NN-DRL method (median: 503.2 s, interquartile range: 398.6–620.4 s). The box of the DDG method is completely below that of the NN-DRL method, visually demonstrating its efficiency advantage. The average task completion time of the DDG method (381.2±19.8 s) is 87.7 seconds shorter than that of the NN-DRL method (468.9±23.5 s), representing a relative improvement of 18.7%. This improvement is reflected in the box plot as a clear vertical offset. The interquartile range of the DDG method (36.6 $s$) is narrower than that of the NN-DRL method (41.6 $s$), indicating lower sensitivity to different scenario configurations and better predictability in time. The GA-DZ method has valid data only in 6 successful scenarios (median: 492.4 s), but considering its 80% failure rate, its average task completion time is effectively infinite.

The relevant performance metrics from the 30 randomly generated simulation scenarios are summarized in Table 6. Based on the comprehensive statistical analysis across 30 randomized scenarios, the proposed DDG method demonstrates significant and consistent advantages over the benchmark methods [6,9]. While maintaining statistically equivalent safety

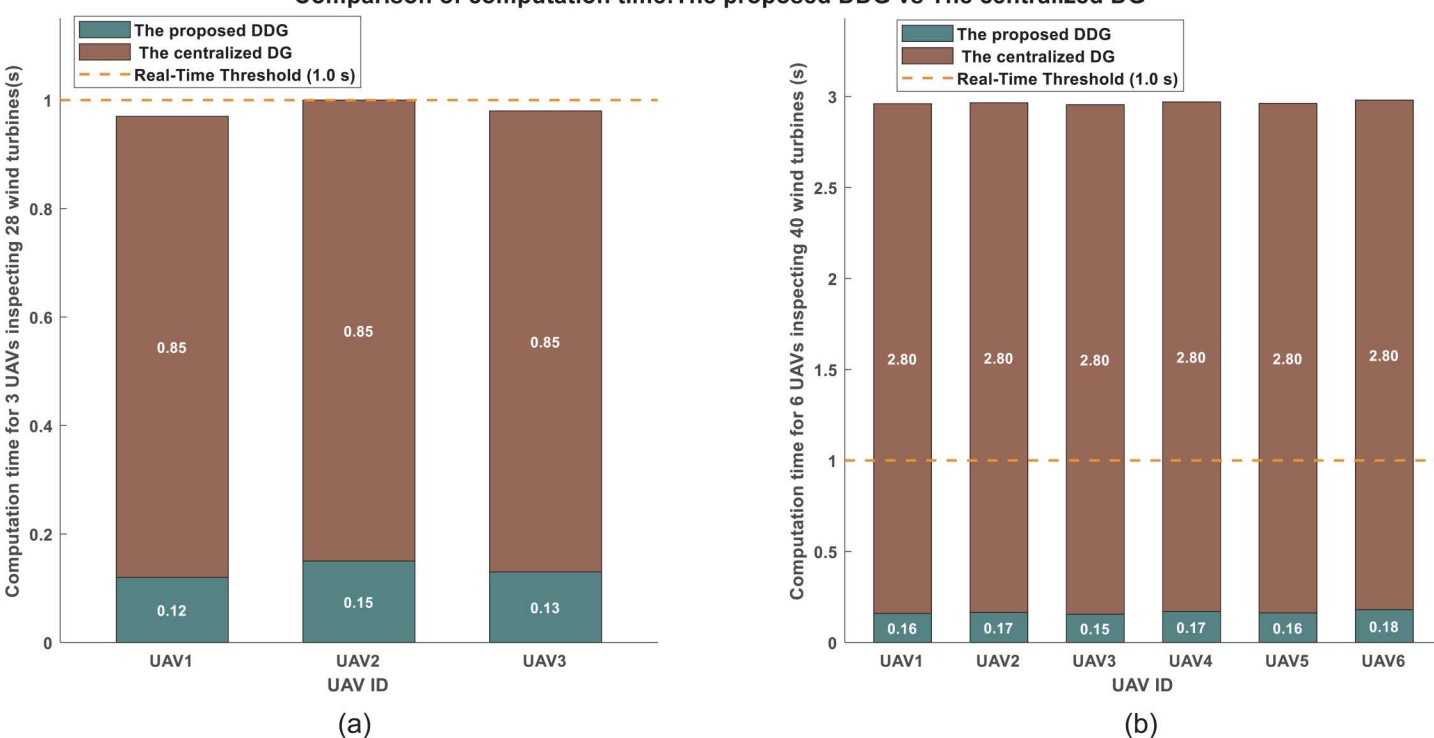

Fig 10.  Comparison of computation time. (a) The comparison of computation time for 3 UAVs inspecting 28 wind turbines (b) The comparison of computation time for 6 UAVs inspecting 40 wind turbines.

Table 6.  The comprehensive statistical results using the three methods.

| Performance metric | Proposed DDG method | GA-DZ Method [6] | NN-DRL Method [9] |
|---|---|---|---|
| Success rate | 100% (30/30) | 20% (6/30) | 100% (30/30) |
| Minimum Safety Distance ($m$) | $5.36 \pm 0.29$ | $4.13 \pm 0.25$ | $5.25 \pm 0.42$ |
| Task Completion Time ($s$) | $381.2 \pm 19.8$ | $\infty$ (80% failure) | $468.9 \pm 23.5$ |

performance ($5.36 \pm 0.29$ m) and perfect reliability (100% success rate) compared to the NN-DRL method, it achieves a remarkable 18.7% reduction in task completion time (381.2 vs. 468.9 seconds). This combination of enhanced efficiency and unwavering reliability—standing in sharp contrast to the GA-DZ method, which fails in 80% of scenarios due to collisions or energy exhaustion—establishes the DDG framework as a highly efficient and robust solution for practical offshore wind farm inspection, particularly well-suited for resource-constrained UAV platforms.

In summary, by incorporating collision avoidance constraints, trajectory optimization, and maximum range constraints, the proposed method enables the multi-UAVs to ensure operational safety while reducing the overall inspection time during offshore wind turbine inspections.

## Scalability analysis

To concretely address scalability and computational burden, we performed a head-to-head comparison between our distributed DDG and an equivalent centralized game-theoretic solver. The centralized solver uses the same PMP principle

and the proposed cost structure but optimizes the trajectories of all UAVs simultaneously using global information. The results are presented in the Fig 11.

1) For the 3-UAV / 28-turbine scenario

The proposed DDG: Per-UAV computation times are 0.12s, 0.15s, and 0.13s (average: $0.133s$).

The centralized solver: The joint optimization requires 0.85s to compute a solution for the entire system. Therefore, the distributed system is $6.5$ times faster per planning cycle when considering parallel execution, which is shown in the Fig 11(a).

2) For the 6-UAV / 40-turbine scenario:

The proposed DDG: Per-UAV times range from 0.155s to 0.18s (average: $0.165s$).

The centralized solver: Computation time surges to $2.8s$.

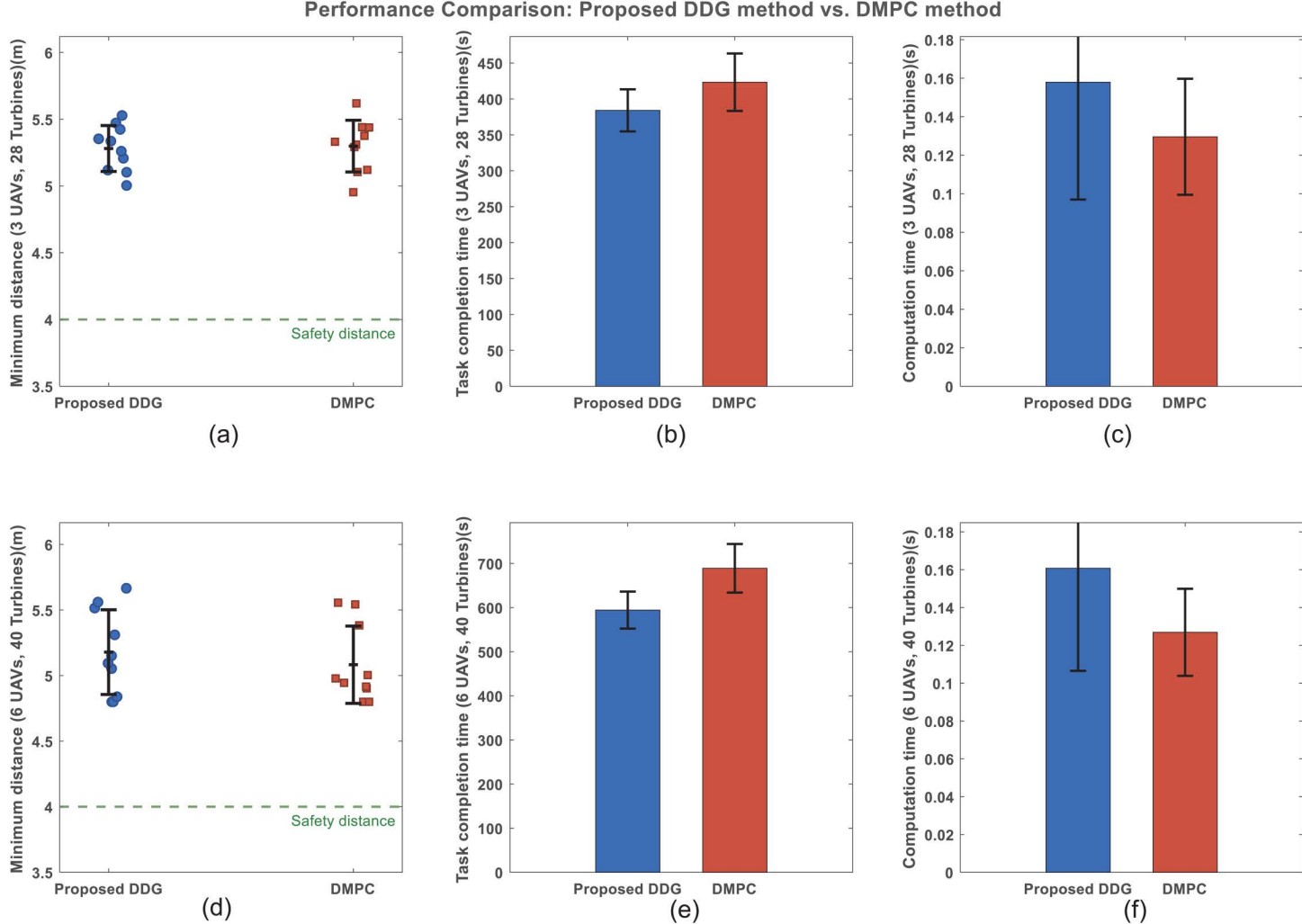

**Fig 11. Performance Comparison.** **(a)** The minimum distance for 3 UAVs inspecting 28 wind turbines with the two methods **(b)** The task completion for 3 UAVs inspecting 28 wind turbines with the two methods **(c)** The computation time for 3 UAVs inspecting 28 wind turbines with the two methods **(d)** The minimum distance for for 6 UAVs inspecting 40 wind turbines with the two methods **(e)** The task completion for 6 UAVs inspecting 40 wind turbines with the two methods **(f)** The computation time for 6 UAVs inspecting 40 wind turbines with the two methods.

Then, the performance gap widens significantly. The distributed system is now 17 times faster on average, and critically, the centralized time far exceeds the $1.0s$ real-time threshold, which is shown in the Fig 11(b). This comparative analysis provides direct, empirical evidence that our proposed DDG framework successfully avoids the combinatorial explosion typical of centralized optimal control.

Notably, when the fleet size was increased by 100% (from 3 to 6 UAVs), the average per-agent computation time of our DDG method increased by only about 24%. This sub-linear growth is a direct outcome of the distributed architecture, where each UAV solves a local problem whose complexity is bounded by the size of its neighborhood $|\mathcal{N}_i|$, rather than by the global fleet size $N$.

To further demonstrate the advantages of the proposed DDG method, a comparative analysis was conducted against the widely-used DMPC approach [15] under two operational paradigms: 3 UAVs inspecting 28 wind turbines and 6 UAVs inspecting 40 wind turbines. For each paradigm, 10 different random simulation scenarios were generated. The following performance metrics were statistically evaluated: the minimum safe distance between UAVs and obstacles (including turbines and other UAVs within sensing range), the total task completion time, and the on-board computation time. In the DMPC [15] implementation, the prediction horizon is set to 10 steps. To ensure a fair comparison between the algorithms, the cost function is kept consistent with that of the DDG approach, and the fmincon solver is employed for optimization. The statistical results are presented in Fig 12.

Fig 12(a) and Fig 12(d) illustrate that, in terms of safety, both methods successfully avoid collisions in all simulations, consistently maintaining obstacle-avoidance distances above the $4.0m$ safety threshold. This confirms that the DDG method preserves a safety level comparable to that of DMPC.

In terms of task efficiency, as observed in Fig 12(b) and Fig 12(e), the DDG method reduces the average task completion time by approximately 12%. For instance, in the 6-UAV scenario, DDG achieves $580 \pm 70s$, whereas DMPC [15] requires $659 \pm 79s$. This improvement stems from the game-theoretic foundation of DDG, which explicitly models the strategic interactions among agents and drives the system toward a NE. Under the assumption of strong connectivity, the L-NE attained by DDG is guaranteed to be globally optimal, ensuring that each UAV's trajectory is globally balanced and coordinated. In contrast, DMPC [15] relies on algorithmic optimization to seek locally optimal solutions at each sampling instant. While such solutions may optimize certain global objectives, they often do so at the expense of individual agent performance, leading to longer overall mission times.

Regarding computational efficiency, Fig 12(c) and Fig 12(f) demonstrate that DDG exceeds the on-board computation time of DMPC by 1.9% to 34.3%. In the 6-UAV scenario, DDG requires $0.165 \pm 0.650s$, whereas DMPC only demands $0.13 \pm 0.840s$. This significant difference stems from the fact that DDG needs to obtain the L-NE strategy through online game theory at each sampling instant. In contrast, DMPC solve an online optimization problem at every sampling interval, the prediction horizon interval of DMPC is shorter than the game-theoretic cycle of DDG at each sampling time, while DDG exhibits only a mild increase in computation, underscoring its superior scalability and real-time capability.

In summary, the proposed DDG method not only matches DMPC in safety assurance but also significantly outperforms it in task efficiency, even though its online computation time is slightly higher. Its game-theoretic formulation ensures globally balanced trajectories with low communication and computation demands, surpassing both the GA-DZ [6] and NN-DRL methods [9]. This makes it particularly suitable for distributed cooperative inspection in offshore wind farms, where communication is limited.

## Conclusion

This paper introduces an optimal coordinated control strategy designed to minimize task completion time for multi-UAV inspection systems in offshore wind farms, subject to limited sensing capabilities and round-trip mission constraints. The coordination challenge is formulated using an novel DDG framework, which avoids the need for global system information.

The proposed model explicitly integrates round-trip requirements into a game-theoretic objective function to facilitate energy-aware trajectory planning. With a strongly connected communication graph, the L-NE from decentralized solving of the DDG provably converges to the G-NE, thereby ensuring system-wide coordination optimality under energy and operational constraints. Simulation results validate the framework's efficacy, confirming its ability to enhance inspection efficiency through a marked reduction in task completion time.

## Limitations and future work

Current Limitations: The present model assumes ideal, delay-free communication within the sensing range and does not explicitly account for dynamic environmental disturbances such as wind gusts.

To enhance the realism and robustness of the proposed framework, future work will focus on incorporating time-varying communication topologies, developing communication delay compensation mechanisms, modeling dynamic wind fields, and formulating strategies to handle unexpected obstacles. Integrating stochastic wind models into the dynamics and cost function for more resilient trajectory planning. Additionally, large-scale simulations (e.g., $N > 20$) will be conducted to empirically quantify the relationship between system scale and performance, further validating the scalability and practical applicability of the method.

## Supporting information

**S1 Appendix. This appendix contains the detailed derivation of the adjoint system (33).**
(PDF)

## Author contributions

**Funding acquisition:** Houmin Wang, Wenyan Xue.

**Software:** Yunqi Liao, Siming Yu.

**Visualization:** Houmin Wang, Wenyan Xue.

**Writing – original draft:** Yunqi Liao.

**Writing – review & editing:** Yunqi Liao, Shuyuan You.

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
