## [Decision Letter · Decision Letter 0]

2 Dec 2025

Dear Dr. wenyan,

Thank you for submitting your manuscript to PLOS ONE. After careful consideration, we feel that it has merit but does not fully meet PLOS ONE’s publication criteria as it currently stands. Therefore, we invite you to submit a revised version of the manuscript that addresses the points raised during the review process.

We look forward to receiving your revised manuscript.

Kind regards,

Tri-Hai Nguyen, Ph.D.

Academic Editor

PLOS ONE

Journal Requirements:

This work is supported by the funding from the Efficient space-time coordination of swarm aircraft (No. 360302022401), Wenyan Xue, visualization. The Research on Motion Control Mechanism and Regulation Strategy for Brain Computer Interface (No.360302042406), Houmin Wang, visualization.

Thanks to Yunqi Liao, Shuyuan You, Houmin Wang and Siming Yu for their help on this project. This work is supported by the funding from the Efficient space-time coordination of swarm aircraft (No. 360302022401) and the Research on Motion Control Mechanism and Regulation Strategy for Brain Computer Interface (No.360302042406).

This work is supported by the funding from the Efficient space-time coordination of swarm aircraft (No. 360302022401), Wenyan Xue, visualization. The Research on Motion Control Mechanism and Regulation Strategy for Brain Computer Interface (No.360302042406), Houmin Wang, visualization.

5. We note that your Data Availability Statement is currently as follows: All relevant data are within the manuscript and its Supporting Information files.

7. Please amend the manuscript submission data (via Edit Submission) to include author Wenyan Xue

8. Please amend your authorship list in your manuscript file to include author Xue Wen wenyan

9. Please ensure that you refer to Figure 2 in your text as, if accepted, production will need this reference to link the reader to the figure.

Reviewers' comments:

Reviewer's Responses to Questions

**Comments to the Author**

1. Is the manuscript technically sound, and do the data support the conclusions?

Reviewer #1: Yes

Reviewer #2: Yes

Reviewer #3: Yes

2. Has the statistical analysis been performed appropriately and rigorously?

Reviewer #1: Yes

Reviewer #2: Yes

Reviewer #3: No

3. Have the authors made all data underlying the findings in their manuscript fully available?

Reviewer #1: Yes

Reviewer #2: Yes

Reviewer #3: No

4. Is the manuscript presented in an intelligible fashion and written in standard English?

Reviewer #1: Yes

Reviewer #2: Yes

Reviewer #3: Yes

Reviewer #1: TITLE: A distributed differential game approach to trajectory planning for offshore wind farm inspection

AUTHOR: Wenyan Xue – Guangdong Ocean University, China

OPINION

The manuscript presents a solid and technically well-founded proposal by employing a formulation based on the Distributed Differential Game (DDG) framework for trajectory planning of unmanned aerial vehicles (UAVs) engaged in cooperative inspections of offshore wind farms. The study is both relevant and timely, considering the growing application of autonomous systems in maritime environments and the inherent challenges related to communication limitations and energy consumption in multi-agent systems. The text demonstrates an adequate conceptual command of differential game theory and distributed control, establishing a consistent link between trajectory optimization, energy efficiency, and operational safety.

From a scientific standpoint, the paper exhibits clear merit by proposing an integration between optimal control and game theory, resulting in a decentralized approach that overcomes the limitations of traditional centralized methods. The theoretical framework is robust, and the demonstration of convergence from the local Nash equilibrium (L-NE) to the global Nash equilibrium (G-NE) is logically developed. However, the mathematical exposition is dense, and certain sections could benefit from a more didactic presentation—particularly those introducing second-order differential equations and the modelling of the multi-UAV system. A summary table consolidating the variables and parameters used would facilitate readability and enhance the reproducibility of the model.

The introduction adequately covers the state of the art, but the comparative discussion with deep reinforcement learning (NN-DRL) and genetic optimization (GA-DZ) methods could be expanded to highlight the incremental originality of the proposed approach more clearly. The simulations are relevant and effectively illustrate trajectory behaviour, although figure captions could be refined for greater interpretive clarity. It would also be desirable to include a more detailed analysis of the computational cost of the proposed solution and an assessment of the model’s scalability as the number of agents increases—an essential aspect in distributed systems.

From a theoretical perspective, the proposed formulation is consistent and aligns with the principles of multi-objective optimization applied to cooperative dynamic systems. The demonstration of convergence toward the G-NE, based on the positive definiteness of the Ω matrix, is well-structured, but would benefit from a more rigorous explanation of the relationship between communication graph connectivity and the uniqueness of the solution. Similarly, practical aspects such as communication delays, wind variation, and possible link losses could be discussed as limitations or directions for future work, which would add both realism and practical depth to the proposed model.

Overall, the article demonstrates high technical quality, mathematical rigour, and strong potential impact in the field of autonomous systems and control engineering. The proposal is innovative in integrating energy constraints, return-to-base requirements, and multi-agent cooperation under a unified theoretical framework. The contributions are significant and can serve as a valuable reference for future research on intelligent inspection of offshore wind turbines.

Reviewer #2: The manuscript introduces a distributed differential game (DDG) framework for optimal trajectory planning of multiple UAVs inspecting offshore wind farms. It addresses critical issues like energy constraints, local communication, and round-trip guarantees. The theoretical analysis includes convergence to global Nash equilibrium (G-NE), supported by simulations comparing the proposed method against GA-DZ and NN-DRL baselines.

Comments

• The GA-DZ and NN-DRL benchmarks lack implementation details. There is no mention of training setup, DRL architecture, or hyperparameter tuning. Fairness of comparison is unclear.

• The system is only tested on 3 UAVs. There is no analysis on how the framework performs with larger teams, which is critical for offshore inspection scalability.

• The convergence from L-NE to G-NE assumes a positive definite matrix Ω but does not explain how it is constructed or verified in practice. This weakens the rigor of the theoretical claim.

• The method involves solving a two-point boundary value problem for each UAV, but computational complexity or real-time feasibility is not discussed.

• Lack of Reproducibility

Simulation settings, UAV parameters, cost matrices (Q, R, Rij), and obstacle configurations are not fully specified.

Minor Points

• Language is mostly clear, but the manuscript needs polishing for grammar (e.g., "This model do not..." → "This model does not...").

• Figures lack proper resolution and labels (especially Fig. 3 and 4).

• Several variables are introduced before being defined. Maintain clear notation flow.

• References are current but rely heavily on the authors’ previous work; diversify citations.

Reviewer #3: The paper tackles the problem of planning trajectories for a team of drones inspecting offshore wind turbines. The main challenges are: limited communication, avoiding collisions, and finite battery life, with a specific emphasis on making sure the drones can make it back to base. The authors frame this as a differential game where each drone is a player making decisions based only on local information from its neighbors. The core idea is to find a Nash Equilibrium for this game, and they make a theoretical argument that the local equilibrium found by their distributed method actually converges to a global one for the whole system. They compare their method against two others from the literature (a genetic algorithm approach and a neural network/deep RL approach) in simulation and show that their method is better at completing the inspection quickly and, crucially, ensures all drones return safely.

-The use of a distributed differential game is a sophisticated and appropriate framework for this kind of multi-agent, conflicting-objectives problem. It is a step up from more basic optimization or learning techniques.

-The aspiration to prove convergence from a local to a global Nash equilibrium is commendable and, if solidly demonstrated, would be a valuable theoretical contribution.

-The simulation results, as presented, do seem to show a clear advantage over the chosen benchmarks, particularly in guaranteeing safety and complete mission success.

--- MAJOR CONCERNS ---

My main concerns revolve around the technical details, which are currently presented in a way that makes it hard to fully evaluate the claims.

1. The Proof of Proposition 1 (Convergence of L-NE to G-NE). This is the theoretical heart of the paper, and I am afraid it is currently not convincing. The proof feels more like a sketch. My main confusion is with the matrix \Omega. It appears in equation (25) seemingly out of nowhere, and its connection to the original Hamiltonian and the system dynamics is not explained. The entire argument hinges on \Omega being positive definite, but we are given no reason to believe this is true. The subsequent steps, which link the adjoint system's uniqueness to the global equilibrium, are stated but not rigorously derived. For the paper to fulfill its theoretical promise, this proof needs to be substantially expanded and clarified.

2. Numerical Solution of the Two-Point Boundary Value Problem. The paper is almost completely silent on how the rather complex optimal control problem is actually solved. Equations (20) and (21) describe a TPBVP that is notoriously difficult to solve numerically, especially for a nonlinear, multi-agent system. The manuscript gives no information on the numerical methods used. The absence of these details makes it impossible to reproduce the results and leaves a major gap in the methodology. It also raises the question of computational burden, which is an important practical consideration for real-time application.

3. Clarity of Presentation and Figures. The presentation significantly hampers understanding.

-The schematic in Figure 1 is too small. The caption does not explain what the blocks represent or how information flows between them. A more detailed diagram and a self-explanatory caption are essential.

-In Figure 3, it's hard to see why one inspection trajectory is better than another.

-In Figure 4, It is not clear how to evaluate the curves and how to compare them among the different methods. Only in (b) it seems that the green line have failed.

-The description of the simulation setup is sparse. Communication range, how the task allocation was done initially, and the precise parameters for the benchmarks are all missing details that would help in judging the fairness of the comparison.

--- MINOR POINTS ---

-The abstract says "enhances inspection efficiency and reducing task completion time" - it should be "reduces".

-Table 1 is a bit strange and feels unnecessary. Also, reporting the mean convergence time for one successful run is not very statistically meaningful. It would be stronger to see results over multiple randomized trials.

-Are there other modern distributed control methods, such as distributed Model Predictive Control, that could be used for a more comprehensive comparison?

--- CONCLUSION ---

In summary, the current manuscript has significant gaps in its theoretical justification and methodological transparency that prevent me from recommending it for publication in its present form.

The authors are on to something good, but they need to do the hard work of fleshing out the proofs and providing the missing algorithmic details. The presentation, especially the figures, needs a major overhaul to communicate the results effectively. I would be very interested in seeing a revised version that addresses these concerns.

**Do you want your identity to be public for this peer review?** For information about this choice, including consent withdrawal, please see our Privacy Policy

Reviewer #1: **Yes:** Miércio Cardoso de Alcântara Neto

Reviewer #2: **Yes:** Asaad Ahmed Gad-Elrab

Reviewer #3: No

---

## [Author Response · Author response to Decision Letter 1]

11 Jan 2026

Thank you very much for your recent e-mail concerning the first submission of the paper. We sincerely thank you and the reviewers for the time spent and effort in reviewing this paper. We have studied the comments carefully and have made significant improvements according to the valuable comments and suggestions of the editor and reviewers. We believe after these modifications, the quality of the revised manuscript can meet the requirements of honorary journals. The detailed revisions are marked in red color in the revised paper. Responses to each reviewer’s comments are enclosed in a separate file labeled 'Response to Reviewers'.

---

## [Decision Letter · Decision Letter 1]

2 Mar 2026

A distributed differential game approach to trajectory planning for offshore wind farm inspection

PONE-D-25-54513R1

Dear Dr. Xue,

We’re pleased to inform you that your manuscript has been judged scientifically suitable for publication and will be formally accepted for publication once it meets all outstanding technical requirements.

Kind regards,

Tri-Hai Nguyen, Ph.D.

Academic Editor

PLOS One

Additional Editor Comments (optional):

The authors have addressed the comments from reviewers.

Reviewers' comments:

Reviewer's Responses to Questions

**Comments to the Author**

Reviewer #1: All comments have been addressed

2. Is the manuscript technically sound, and do the data support the conclusions?

Reviewer #1: Yes

3. Has the statistical analysis been performed appropriately and rigorously?

Reviewer #1: Yes

4. Have the authors made all data underlying the findings in their manuscript fully available?

Reviewer #1: Yes

5. Is the manuscript presented in an intelligible fashion and written in standard English?

Reviewer #1: Yes

Reviewer #1: All of my comments were taken into consideration, and I find the current version to be satisfactory.

**Do you want your identity to be public for this peer review?** For information about this choice, including consent withdrawal, please see our Privacy Policy

Reviewer #1: **Yes:** Prof. Miércio Cardoso de Alcântara Neto, DSc

---

## [Editor Report · Acceptance letter]

PONE-D-25-54513R1

PLOS One

Dear Dr. Xue,

I'm pleased to inform you that your manuscript has been deemed suitable for publication in PLOS One. Congratulations! Your manuscript is now being handed over to our production team.

Kind regards,

on behalf of

Dr. Tri-Hai Nguyen

Academic Editor

PLOS One